# MaxGap Bandit: Adaptive Algorithms for Approximate Ranking

**Sumeet Katariya** *
UW-Madison and Amazon
sumeetsk@gmail.com

**Ardhendu Tripathy** *
UW-Madison
astripathy@wisc.edu

**Robert Nowak**
UW-Madison
rdnowak@wisc.edu

## Abstract

This paper studies the problem of adaptively sampling from $K$ distributions (arms) in order to identify the largest gap between any two adjacent means. We call this the MaxGap-bandit problem. This problem arises naturally in approximate ranking, noisy sorting, outlier detection, and top-arm identification in bandits. The key novelty of the MaxGap bandit problem is that it aims to adaptively determine the natural partitioning of the distributions into a subset with larger means and a subset with smaller means, where the split is determined by the largest gap rather than a pre-specified rank or threshold. Estimating an arm's gap requires sampling its neighboring arms in addition to itself, and this dependence results in a novel hardness parameter that characterizes the sample complexity of the problem. We propose elimination and UCB-style algorithms and show that they are minimax optimal. Our experiments show that the UCB-style algorithms require 6-8x fewer samples than non-adaptive sampling to achieve the same error.

## 1 Introduction

Consider an algorithm that can draw i.i.d. samples from $K$ unknown distributions. The goal is to partially rank the distributions according to their (unknown) means. This model encompasses many problems including best-arms identification (BAI) in multi-armed bandits, noisy sorting and ranking, and outlier detection. Partial ranking is often preferred to complete ranking because correctly ordering distributions with nearly equal means is an expensive task (in terms of number of required samples). Moreover, in many applications it is arguably unnecessary to resolve the order of such close distributions. This observation motivates algorithms that aim to recover a partial ordering of groups of distributions having similar means. This entails identifying large "gaps" in the ordered sequence of means. The focus of this paper is the fundamental problem of finding the *largest gap* by sampling adaptively. Identification of the largest gap separates the distributions into two groups, and recursive application can identify any desired number of groupings in a partial order.

As an illustration, consider a subset of images from the Chicago streetview dataset [17] shown in Fig. 1. In this study, people were asked to judge how safe each scene looks [18], and a larger mean indicates a safer looking scene. While each person has a different sense of how safe an image looks, when aggregated there are clear trends in the safety scores (denoted by $\mu_{(i)}$) of the images. Fig. 1 schematically shows the distribution of scores given by people as a bell curve below each image. Assuming the sample means are close to their true means, one can nominally classify them as 'safe', 'maybe unsafe' and 'unsafe' as indicated in Fig. 1. Here we have implicitly used the large gaps $\mu_{(2)} - \mu_{(3)}$ and $\mu_{(4)} - \mu_{(5)}$ to mark the boundaries. Note that finding the safest image (BAI) is hard as we need a lot of human responses to decide the larger mean between the two rightmost distributions; it is also arguably unnecessary. A common way to address this problem is to specify a tolerance $\epsilon$ [7],

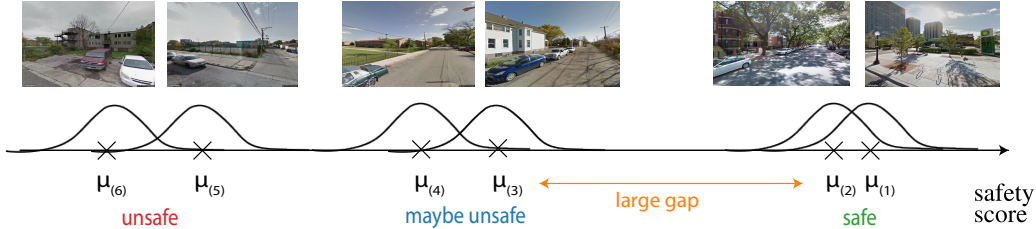

Figure 1: Six representative images from Chicago streetview dataset and their safety (Borda) scores.

and stop sampling if the means are less than $\epsilon$ apart; however determining this can require $\Omega(1/\epsilon^2)$ samples. Distinguishing the top 2 distributions from the rest is easy and can be efficiently done using top-$m$ arm identification [15], however this requires the experimenter to prescribe the location $m = 2$ where a large gap exists which is unknown. *Automatically identifying natural splits in the set of distributions is the aim of the new theory and algorithms we propose. We call this problem of adaptive sampling to find the largest gap the MaxGap-bandit problem.*

## 1.1 Notation and Problem Statement

We will use multi-armed bandit terminology and notation throughout the paper. The $K$ distributions will be called *arms* and drawing a sample from a distribution will be refered to as *sampling the arm*. Let $\mu_i \in \mathbb{R}$ denote the mean of the $i$-th arm, $i \in \{1, 2, \ldots, K\} =: [K]$. We add a parenthesis around the subscript $j$ to indicate the $j$-th largest mean, i.e., $\mu_{(K)} \leq \mu_{(K-1)} \leq \cdots \leq \mu_{(1)}$. For the $i$-th arm, we define its gap $\Delta_i$ to be the maximum of its left and right gaps, i.e.,

$$\Delta_i = \max\{\mu_{(\ell)} - \mu_{(\ell+1)}, \mu_{(\ell-1)} - \mu_{(\ell)}\} \quad \text{where } \mu_i = \mu_{(\ell)}. \tag{1}$$

We define $\mu_{(0)} = -\infty$ and $\mu_{(K+1)} = \infty$ to account for the fact that extreme arms have only one gap. The goal of the MaxGap-bandit problem is to (adaptively) sample the arms and return two clusters

$$C_1 = \{(1), (2), \ldots, (m)\} \quad \text{and} \quad C_2 = \{(m+1), \ldots, (K)\},$$

where $m$ is the rank of the arm with the largest gap between *adjacent* means, i.e.,

$$m = \underset{j \in [K-1]}{\arg\max} \; \mu_{(j)} - \mu_{(j+1)}. \tag{2}$$

The mean values are unknown as is the ordering of the arms according to their means. A solution to the MaxGap-bandit problem is an algorithm which given a probability of error $\delta > 0$, samples the arms and upon stopping partitions $[K]$ into two clusters $\widehat{C}_1$ and $\widehat{C}_2$ such that

$$\mathbb{P}(\widehat{C}_1 \neq C_1) \leq \delta. \tag{3}$$

This setting is known as the fixed-confidence setting [10], and the goal is to achieve the probably correct clustering using as few samples as possible. In the sequel, we assume that $m$ is uniquely defined and let $\Delta_{\max} = \Delta_{i^*}$ where $\mu_{i^*} = \mu_{(m)}$.

## 1.2 Comparison to a Naive Algorithm: Sort then search for MaxGap

The MaxGap-bandit problem is not equivalent to BAI on $\binom{K}{2}$ gaps since the MaxGap-bandit problem requires identifying the largest gap between *adjacent* arm means (BAI on $\binom{K}{2}$ gaps would always identify $\mu_{(1)} - \mu_{(K)}$ as the largest gap). This suggests a naive two-step algorithm: we first sample the arms enough number of times so as to identify all pairs of adjacent arms (i.e., we sort the arms according to their means), and then run a BAI bandit algorithm [13] on the $(K-1)$ gaps between adjacent arms to identify the largest gap (an unbiased sample of the gap can be obtained by taking the difference of the samples of the two arms forming the gap).

We analyze the sample complexity of this naive algorithm in Appendix A , and discuss the results here for an example. Consider the arrangement of means shown in Fig. 2 where there is one large gap $\Delta_{\max}$ and all the other gaps are equal to $\Delta_{\min} \ll \Delta_{\max}$. The naive algorithm's sample complexity is $\Omega(K/\Delta_{\min}^2)$, as the first sorting step requires these many samples, which can be very large.

Is this sorting of the arm means necessary? For instance, we do not need to sort $K$ real numbers in order to cluster them according to the largest gap [1].

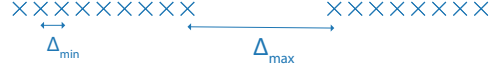

The algorithms we propose in this paper solve the MaxGap-bandit problem without necessarily sorting

Figure 2: Arm means with one large gap.

the arm means. For the configuration in Fig. 2 they require $\tilde{O}(K/\Delta_{\max}^2)$ samples, giving a saving of approximately $(\Delta_{\max}/\Delta_{\min})^2$ samples.

The analysis of our algorithms suggests a novel hardness parameter for the MaxGap-bandit problem that we discuss next. We let $\Delta_{i,j} := \mu_j - \mu_i$ for all $i, j \in [K]$. We show in Section 5 that the number of samples taken from distribution $i$ due to its right gap is inversely proportional to the square of

$$\gamma_i^r := \max_{j:\Delta_{i,j}>0} \min\left\{\Delta_{i,j}, \Delta_{\max} - \Delta_{i,j}\right\}. \tag{4}$$

For the left gap of $i$ we define $\gamma_i^l$ analogously. The total number of samples drawn from distribution $i$ is inversely proportional to the square of $\gamma_i := \min\{\gamma_i^r, \gamma_i^l\}$. The intuition for Eq. (4) is that distribution $i$ can be eliminated quickly if there is another distribution $j$ that has a moderately large gap from $i$ (so that this gap can be quickly detected), but not too large (so that the gap is easy to distinguish from $\Delta_{\max}$), and (4) chooses the best $j$. We discuss (4) in detail in Section 5, where we show that our algorithms use $O\left(\sum_{i\in[K]/\{(m),(m+1)\}} \gamma_i^{-2} \log(K/\delta\gamma_i)\right)$ samples to find the largest gap with probability at least $1 - \delta$. This sample complexity is minimax optimal.

## 1.3   Summary of Main Results and Paper Organization

In addition to motivating and formulating the MaxGap-bandit problem, we make the following contributions. First, we design elimination and UCB-style algorithms as solutions to the MaxGap-bandit problem that do not require sorting the arm means (Section 3). These algorithms require computing upper bounds on the gaps $\Delta_i$, which can be formulated as a mixed integer optimization problem. We design a computationally efficient dynamic programming subroutine to solve this optimization problem and this is our second contribution (Section 4). Third, we analyze the sample complexity of our proposed algorithms, and discover a novel problem-hardness parameter (Section 5). This parameter arises because of the arm interactions in the MaxGap-bandit problem where, in order to reduce uncertainty in the value of an arm's gap, we not only need to sample the said arm but also its neighboring arms. Fourth, we show that this sample complexity is minimax optimal (Section 6). Finally, we evaluate the empirical performance of our algorithms on simulated and real datasets and observe that they require 6-8x fewer samples than non-adaptive sampling to achieve the same error (Section 7).

## 2   Related Work

One line of related research is best-arm identification (BAI) in multi-armed bandits. A typical goal in this setting is to identify the top-$m$ arms with largest means, where $m$ is a *prespecified* number [15, 16, 1, 3, 9, 4, 14, 7, 20]. As explained in Section 1, our motivation behind formulating the MaxGap-bandit problem is to have an adaptive algorithm which finds the "natural" set of top arms as delineated by the largest gap in consecutive mean values. Our work can also be used to automatically detect "outlier" arms [23].

The MaxGap-bandit problem is different from the standard multi-armed bandit because of the local dependence of an arm's gap on other arms. Other best-arm settings where an arm's reward can inform the quality of other arms include linear bandits [22] and combinatorial bandits [5, 11]. In these problems, the decision space is known to the learner, i.e., the vectors corresponding to the arms in linear bandits and the subsets of arms over which the objective function is to be optimized in combinatorial bandits is known to the learner. However in our problem, we do not know the sorted order of the arm means, i.e., the set of all valid gaps is unknown *a priori*. Our problem does not reduce to these settings.

Another related problem is noisy sorting and ranking. Here the typical goal is to sort a list using noisy pairwise comparisons. Our framework encompasses noisy ranking based on Borda scores [1]. The Borda score of an item is the probability that it is ranked higher in a pairwise comparison with another item chosen uniformly at random. In our setting, the Borda score is the mean of each distribution. Much of the theoretical computer science literature on this topic assumes a bounded noise model for comparisons (i.e., comparisons are probably correct with a positive margin) [8, 6, 2, 21]. This is unrealistic in many real-world applications since near equals or outright ties are not uncommon. The largest gap problem we study can be used to (partially) order items into two natural groups, one with large means and one with small means. Previous related work considered a similar problem with prescribed (non-adaptive) quantile groupings [18].

## 3 MaxGap Bandit Algorithms

We propose elimination [7] and UCB [13] style algorithms for the MaxGap-bandit problem. These algorithms operate on the arm *gaps* instead of the arm *means*. The subroutine to construct confidence intervals on the gaps (denoted by $\text{U}\Delta_a(t)$) using confidence intervals on the arm means (denoted by $[l_a(t), r_a(t)]$) is described in Algorithm 4 in Section 4, and this subroutine is used by all three algorithms described in this section.

### 3.1 Elimination Algorithm: `MaxGapElim`

At each time step, `MaxGapElim` (Algorithm 1) samples all arms in an active set consisting of arms $a$ whose gap upper bound $\text{U}\Delta_a$ is larger than the global lower bound $\text{L}\Delta$ on the maximum gap, and stops when there are only two arms in the active set.

---

**Algorithm 1** `MaxGapElim`

---

1: Initialize active set $A = [K]$
2: **for** $t = 1, 2, \ldots$ **do**            *// rounds*
3:    $\forall\, a \in A$, sample arm $a$, compute $[l_a(t), r_a(t)]$ using (5).     *//arm confidence intervals*
4:    $\forall\, a \in A$, compute $\text{U}\Delta_a(t)$ using Algorithm 4.     *// upper bound on arm max gap*
5:    Compute $\text{L}\Delta(t)$ using (9).     *// lower bound on max gap*
6:    $\forall\, a \in A$, if $\text{U}\Delta_a(t) \leq \text{L}\Delta(t)$, $A = A \setminus a$.     *// Elimination*
7:    If $|A| = 2$, stop. Return clusters using max gap in the empirical means. *// Stopping condition*

---

### 3.2 UCB algorithms: `MaxGapUCB` and `MaxGapTop2UCB`

`MaxGapUCB` (Algorithm 2) is motivated from the principle of "optimism in the face of uncertainty". It samples *all* arms with the highest gap upper bound. Note that there are at least two arms with the highest gap upper bound because any gap is shared by at least two arms (one on the right and one on the left). The stopping condition is akin to the stopping condition in Jamieson et al. [13].

---

**Algorithm 2** `MaxGapUCB`

---

1: Initialize $\mathcal{U} = [K]$.
2: **for** $t = 1, 2, \ldots$ **do**
3:    $\forall a \in \mathcal{U}$, sample $a$ and update $[l_a(t), r_a(t)]$ using (5).
4:    $\forall a \in [K]$, compute $\text{U}\Delta_a(t)$ using Algorithm 4.
5:    Let $M_1(t) = \max_{j \in [K]} \text{U}\Delta_j(t)$. Set $\mathcal{U} = \{a : \text{U}\Delta_a(t) = M_1(t)\}$.    *// highest gap-UCB arms*
6:    If $\exists\, i, j$ such that $T_i(t) + T_j(t) \geq c \sum_{a \notin \{i,j\}} T_a(t)$, stop.    *// stopping condition*

---

Alternatively, we can use an LUCB [16]-type algorithm that samples arms which have the two highest gap upper bounds, and stops when the second-largest gap upper bound is smaller than the global lower bound $\text{L}\Delta(t)$ . We refer to this algorithm as `MaxGapTop2UCB` (Algorithm 3).

---

**Algorithm 3** `MaxGapTop2UCB`

---
1: Initialize $\mathcal{U}_1 \cup \mathcal{U}_2 = [K]$.
2: **for** $t = 1, 2, \dots$ **do**
3:    $\forall a \in \mathcal{U}_1 \cup \mathcal{U}_2$, sample $a$ and update $[l_a(t), r_a(t)]$ using (5).
4:    $\forall a \in [K]$, compute $\text{U}\Delta_a(t)$ using Algorithm 4.
5:    Let $M_1(t) = \max_{j \in [K]} \text{U}\Delta_j(t)$. Set $\mathcal{U}_1 = \{a : \text{U}\Delta_a(t) = M_1(t)\}$.   *// highest gap-UCB arms*
6:    Let $M_2(t) = \max_{j \in [K] \setminus \mathcal{U}_1} \text{U}\Delta_j(t)$. Set $\mathcal{U}_2 = \{a : \text{U}\Delta_a(t) = M_2(t)\}$. *// 2nd highest gap-UCB*
7:    Compute $\text{L}\Delta(t)$ using (9). If $M_2(t) < \text{L}\Delta(t)$, stop.

---

---

**Algorithm 4** Procedure to find $\text{U}\Delta_a(t)$

---
1: Set $P_a^r = \{i : l_i(t) \in [l_a(t), r_a(t)]\}$.
2: $\text{U}\Delta_a^r(t) = \max\limits_{i \in P_a^r} \{G_a^r(l_i(t), t)\}$, where $G_a^r(x, t)$ is given by (7).           *// eqn. (8)*
3: Set $P_a^l = \{i : r_i(t) \in [l_a(t), r_a(t)]\}$.
4: $\text{U}\Delta_a^l(t) = \max\limits_{i \in P_a^l} \{G_a^l(r_j(t), t)\}$, where $G_a^l(x, t)$ is given by (19).           *// eqn. (20)*
5: **return** $\text{U}\Delta_a(t) \leftarrow \max\{\text{U}\Delta_a^r(t), \text{U}\Delta_a^l(t)\}$

---

## 4 Confidence Bounds for Gaps

In this section we explain how to construct confidence bounds for the arm gaps (denoted by $\text{U}\Delta_a$ and $\text{L}\Delta$) using confidence bounds for the arm means (denoted by $[l_a, r_a]$). These bounds are key ingredients for the algorithms described in Section 3.

Given i.i.d. samples from arm $a$, an empirical mean $\hat{\mu}_a$ and confidence interval on the arm mean can be constructed using standard methods. Let $T_a(t)$ denote the number of samples from arm $a$ after $t$ time steps of the algorithm. Throughout our analysis and experimentation we use confidence intervals on the mean of the form

$$l_a(t) = \hat{\mu}_a(t) - c_{T_a(t)} \text{ and } r_a(t) = \hat{\mu}_a(t) + c_{T_a(t)}, \text{ where } c_s = \sqrt{\frac{\log(4Ks^2/\delta)}{s}}. \tag{5}$$

The confidence intervals are chosen so that [12]

$$\mathbb{P}(\forall\, t \in \mathbb{N}, \forall\, a \in [K], \mu_a \in [l_a(t), r_a(t)]) \geq 1 - \delta. \tag{6}$$

Conceptually, the confidence intervals on the arm means can be used to construct upper confidence bounds on the mean gaps $\{\Delta_i\}_{i \in [K]}$ in the following manner. Consider all possible configurations of the arm means that satisfy the confidence interval constraints in (5). Each configuration fixes the gaps associated with any arm $a \in [K]$. Then the maximum gap value over all configurations is the upper confidence bound on arm $a$'s gap; we denote it as $\text{U}\Delta_a$. The above procedure can be formulated as a mixed integer linear program (see Appendix B.1). In the algorithms in Section 3, this optimization problem needs to be solved at every time $t$ and for every arm $a \in [K]$ before querying a new sample, which can be practically infeasible. In Algorithm 4, we give an efficient $O(K^2)$ time dynamic programming algorithm to compute $\text{U}\Delta_a$. We next explain the main ideas used in this algorithm, and refer the reader to Appendix B.2 for the proofs.

Each arm $a$ has a right and left gap, $\Delta_a^r := \mu_{(\ell-1)} - \mu_{(\ell)}$ and $\Delta_a^l := \mu_{(\ell)} - \mu_{(\ell+1)}$, where $\ell$ is the rank of $a$, i.e., $\mu_a = \mu_{(\ell)}$. We construct separate upper bounds $\text{U}\Delta_a^r(t)$ and $\text{U}\Delta_a^l(t)$ for these gaps and then define $\text{U}\Delta_a(t) = \max\{\text{U}\Delta_a^r(t), \text{U}\Delta_a^l(t)\}$. Here we provide an intuitive description for how the bounds are computed, focusing on $\text{U}\Delta_a^r(t)$ as an example. To start, suppose the true mean of arm $a$ is known exactly, while the means of other arms are only known to lie within their confidence intervals. If there exist arms that cannot go to the left of arm $a$, one can see that the largest right gap for $a$ is obtained by placing all arms that can go to the left of $a$ at their leftmost positions, and all remaining arms at their rightmost positions, as shown in Fig. 3(a). If however all arms can go to the left of arm $a$, the configuration that gives the largest right gap for $a$ is obtained by placing the arm with the largest upper bound at its right boundary, and all other arms at their left boundaries, as illustrated in Fig. 3(b). We define a function $G_a^r(x, t)$ that takes as input a known position $x$ for the mean of arm $a$

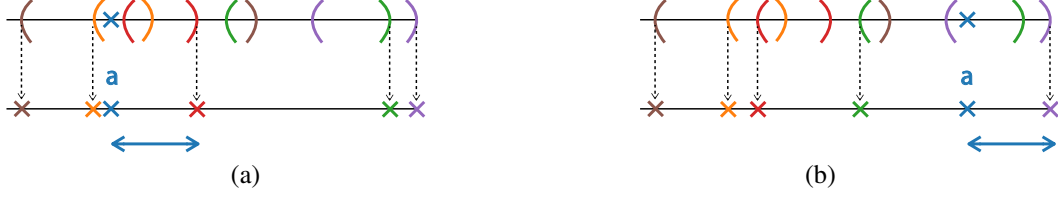

(a)                              (b)

Figure 3: Computing maximum right gap of blue arm when its true mean is known (at position indicated by blue x), while the other means are known only to lie within their confidence intervals. (a) If there exist arms that cannot go to the left of blue (red, green, purple), the largest right gap for blue is obtained by placing all arms that can go to the left of blue at their left boundaries and the remaining arms at their rightmost positions. (b) If all arms can go to the left of blue, the largest right gap for blue is obtained by placing the arm with the largest right confidence bound (purple) at its right boundary and all other arms at their left boundaries.

and the confidence intervals of all other arms at time $t$, and returns the maximum right gap for arm $a$ using the above idea as follows.

$$G_a^r(x,t) = \begin{cases} \min_{j:l_j(t)>x} r_j(t) - x & \text{if } \{j : l_j(t) > x\} \neq \emptyset, \\ \max_{j \neq a} r_j(t) - x & \text{otherwise.} \end{cases} \tag{7}$$

However, the true mean of arm $a$ is not known exactly but only that it lies within its confidence interval. The insight that helps here is that $G_a^r(x,t)$ must achieve its maximum when $x$ is at one of the finite locations in $\{l_j(t) : l_a(t) \leq l_j(t) \leq r_a(t)\}$. We define $P_a^r := \{j : l_a(t) \leq l_j(t) \leq r_a(t)\}$ as the set of arms relevant for the right gap of $a$, and then the maximum possible right gap of $a$ is

$$\text{U}\Delta_a^r(t) = \max\{G_a^r(l_j(t), t) : j \in P_a^r\}. \tag{8}$$

An upper bound for the left gap $\text{U}\Delta_a^l$ can be similarly obtained. We explain this and give a proof of correctness in Appendix B.2.

The algorithms also use a single global lower bound on the maximum gap. To do so, we sort the items according to their empirical means, and find partitions of items that are clearly separated in terms of their confidence intervals. At time $t$, let $(i)_t$ denote the arm with the $i^{\text{th}}$-largest empirical mean, i.e., $\hat{\mu}_{(K)_t}(t) \leq \ldots \hat{\mu}_{(2)_t}(t) \leq \hat{\mu}_{(1)_t}(t)$ (this can be different from the true ranking which is denoted by $(\cdot)$ without the subscript $t$). We *detect* a nonzero gap at arm $k$ if $\max_{a \in \{(k+1)_t, \ldots, (K)_t\}} r_a(t) < \min_{a \in \{(1)_t, \ldots, (k)_t\}} l_a(t)$. Thus, a lower bound on the largest gap is

$$\text{L}\Delta(t) = \max_{k \in [K-1]} \left( \min_{a \in \{(1)_t, \ldots, (k)_t\}} l_a(t) - \max_{a \in \{(k+1)_t, \ldots, (K)_t\}} r_a(t) \right). \tag{9}$$

## 5 Analysis

In this section, we first state the accuracy and sample complexity guarantees for `MaxGapElim` and `MaxGapUCB`, and then discuss our results. The proofs can be found in the Supplementary material.

**Theorem 1.** *With probability* $1 - \delta$, `MaxGapElim`, `MaxGapUCB` *and* `MaxGapTop2UCB` *cluster the arms according to the maximum gap, i.e., they satisfy* (3).

The number of times arm $a$ is sampled by both the algorithms depends on $\gamma_a = \min\{\gamma_a^l, \gamma_a^r\}$ where

$$\gamma_a^r = \max_{j:0<\Delta_{a,j}<\Delta_{\max}} \min\{\Delta_{a,j}, (\Delta_{\max} - \Delta_{a,j})\} \tag{10}$$

$$\gamma_a^l = \max_{j:0<\Delta_{j,a}<\Delta_{\max}} \min\{\Delta_{j,a}, (\Delta_{\max} - \Delta_{j,a})\}, . \tag{11}$$

The maxima is assumed to be $\infty$ in (10) and (11) if there is no $j$ that satisfies the constraint to account for edge arms. The quantity $\gamma_a$ acts as a measure of hardness for arm $a$; Theorem 2 states that `MaxGapElim` and `MaxGapUCB` sample arm $a$ at most $\tilde{O}(1/\gamma_a^2)$ number of times (up to log factors).

**Theorem 2.** *With probability $1 - \delta$, the sample complexity of* `MaxGapElim` *and* `MaxGapUCB` *is bounded by*

$$O\left(\sum_{a \in [K] \setminus \{(m),(m+1)\}} \frac{\log(K/\delta\gamma_a)}{\gamma_a^2}\right)$$

Next, we provide intuition for why the sample complexity depends on the parameters in (10) and (11). In particular, we show that $O((\gamma_a^r)^{-2})$ (resp. $O((\gamma_a^l)^{-2})$) is the number of samples of $a$ required to rule out arm $a$'s right (resp. left) gap from being the largest gap.

Let us focus on the right gap for simplicity. To understand how (10) naturally arises, consider Fig. 4, which denotes the confidence intervals on the means at some time $t$. A lower bound on the gap $\mathtt{L\Delta}(t)$ can be computed between the left and right confidence bounds of arms 10 and 11 respectively as shown. Consider the com-

Figure 4: Arm $a = 7$ is eliminated when a helper arm $j = 4$ is found.

putation of the upper bound $\mathtt{U\Delta}_7^r(t)$ on the right gap of arm $a = 7$. Arm 4 lies to the right of arm 7 with high probability (unlike the arms with dashed confidence intervals), so the upper bound $\mathtt{U\Delta}_7^r(t) \leq r_4(t) - l_7(t)$. Considering only the right gap for simplicity, as soon as $\mathtt{U\Delta}_7^r(t) < \mathtt{L\Delta}(t)$, arm 7 can be eliminated as a candidate for the maximum gap. Thus, an arm $a$ is removed from consideration as soon as we find a *helper* arm $j$ (arm 4 in Fig. 4) that satisfies two properties: (1) the confidence interval of arm $j$ is disjoint from that of arm $a$, and (2) the upper bound $\mathtt{U\Delta}_a^r(t) = r_j(t) - l_a(t) < \mathtt{L\Delta}(t)$. The first of these conditions gives rise to the term $\Delta_{a,j}$ in (10), and the second condition gives rise to the term $(\Delta_{\max} - \Delta_{a,j})$. Since any arm $j$ that satisfies these conditions can act as a helper for arm $a$, we take the maximum over all arms $j$ to yield the smallest sample complexity for arm $a$.

This also shows that if all arms are either very close to $a$ or at a distance approximately $\Delta_{\max}$ from $a$, then the upper bound $\mathtt{U\Delta}_7^r(t) = r_4(t) - l_7(t) > \mathtt{L\Delta}(t)$ and arm 7 cannot be eliminated. Thus arm $a$ could have a small gap with respect to its adjacent arms, but if there is a large gap in the vicinity of arm $a$, it cannot be eliminated quickly. This illustrates that the maximum gap identification problem is not equivalent to best-arm identification (BAI) on gaps. Section 6 formalizes this intuition.

**Key Differences compared to BAI Analysis:** The analysis of `MaxGapUCB` is very different from the standard UCB analysis. On a high-level, in BAI, the number of samples of a sub-optimal arm $i$ is bounded by observing that

$$\text{Arm } i \text{ is pulled} \implies \mu_i + 2c_{T_i(t)} \geq \hat{\mu}_i + c_{T_i(t)} \geq \hat{\mu}_{(1)} + c_{T_{(1)}(t)} \geq \mu_{(1)}$$
$$\implies 2c_{T_i(t)} \geq \mu_{(1)} - \mu_i = \Delta_i. \tag{12}$$

The last inequality *directly* bounds the number of samples $T_i(t)$ of a sub-optimal arm $i$. In `MaxGapUCB`, the gap upper bound is obtained using the confidence intervals of two arms, and the fact that a sub-optimal gap $(i, j)$ has the highest gap-UCB implies that

$$(\mu_j + 2c_{T_j(t)}) - (\mu_i - 2c_{T_i(t)}) \geq (\hat{\mu}_j + c_{T_j(t)}) - (\hat{\mu}_i - 2c_{T_i(t)}) \geq \Delta_{\max}$$
$$\implies 2(c_{T_j(t)} + c_{T_i(t)}) \geq \Delta_{\max} - \Delta_{ij}.$$

Thus unlike the reasoning in (12), the number of samples from arm $i$ is coupled to the number of samples from arm $j$, and $T_i(t) \to \infty$ if $j$ is not sampled enough. We show in our analysis that this cannot happen in `MaxGapUCB`. Furthermore, any arm $i$ is coupled with multiple other arms since the ordering of the arms is unknown, and may have to be sampled even if its own gap is small - a phenomenon absent in standard BAI analysis because of the independence of the arm means. In our proof, we account for all samples of an arm by defining states the arm can belong to (called levels), and arguing about the confidence intervals of the arms in unison.

## 6 Minimax Lower Bound

In this section, we demonstrate that the MaxGap problem is fundamentally different from best-arm identification (BAI) on gaps. We construct a problem instance and prove a lower bound on the number of samples needed by any probably correct algorithm. The lower bound matches the upper bounds in the previous section for this instance.

**Lemma 1.** *Consider a model $\mathcal{B}$ with $K = 4$ normal distributions $\mathcal{P}_i = \mathcal{N}(\mu_i, 1)$, where*

$$\mu_4 = 0, \quad \mu_3 = \epsilon, \quad \mu_2 = \nu + 2\epsilon, \quad \mu_1 = 2\nu + 2\epsilon,$$

*for some $\nu \gg \epsilon > 0$. Then any algorithm that is correct with probability at least $1 - \delta$ must collect $\Omega(1/\epsilon^2)$ samples of arm 4 in expectation.*

*Proof Outline:* The proof uses a standard change of measure argument [10]. We construct another problem instance $\mathcal{B}'$ which has a different maximum gap clustering compared to $\mathcal{B}$ (see Fig. 5, the maxgap clustering in $\mathcal{B}$ is $\{4, 3\} \cup \{2, 1\}$, while the maxgap clustering in $\mathcal{B}'$ is $\{4, 3, 2\} \cup \{1\}$), and show that in order to distinguish between $\mathcal{B}$ and $\mathcal{B}'$, any probably correct algorithm must collect at least $\Omega(1/\epsilon^2)$ samples of arm 4 in expectation (see Appendix E for details). From the definition of $\gamma_a$ using (10),(11), it

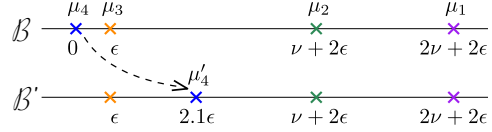

Figure 5: Changing the original bandit model $\mathcal{B}$ to $\mathcal{B}'$. $\mu_4$ is shifted to the right by $2.1\epsilon$. As a result, the maximum gap in $\mathcal{B}'$ is between green and purple.

is easy to check that $\gamma_4 = \epsilon$. Therefore, for problem instance $\mathcal{B}$ our algorithms find the maxgap clustering using at most $O(\log(\epsilon/\delta)/\epsilon^2)$ samples of arm 4 (*c.f.* Theorem 2). This essentially matches the lower bound above.

This example illustrates why the maximum gap identification problem is different from a simple BAI on gaps. Suppose an oracle told a BAI algorithm the ordering of the arm means. Using the ordering it can convert the 4-arm maximum gap problem $\mathcal{B}$ to a BAI problem on 3 *gaps*, with distributions $\mathcal{P}_{4,3} = \mathcal{N}(\epsilon, 2), \mathcal{P}_{3,2} = \mathcal{N}(\nu + \epsilon, 2)$, and $\mathcal{P}_{2,1} = \mathcal{N}(\nu, 2)$. The BAI algorithm can sample arms $i$ and $i + 1$ to get a sample of the gap $(i + 1, i)$. We know from standard BAI analysis [13] that the gap $(4, 3)$ can be eliminated from being the largest by sampling it (and hence arm 4) $O(1/\nu^2)$ times, which can be arbitrarily lower than the $1/\epsilon^2$ lower bound in Lemma 1. Thus the ordering information given to the BAI algorithm is crucial for it to quickly identify the larger gaps. The problem we solve in this paper is identifying the maximum gap when the ordering information is *not* available.

## 7 Experiments

We conduct three experiments. First, we verify the validity of our sample complexity bounds in Section 7.1. We then study the performance of our adaptive algorithms on simulated data in Section 7.2, and on the Streetview dataset in Section 7.3. The code for all experiments is publicly available [19].

### 7.1 Sample Complexity

In Fig. 6(b) and Fig. 6(c), we plot the empirical stopping time against the theoretical sample complexity (Theorem 2) for different arm configurations. We choose the arm configuration in Fig. 6(a) containing $K = 15$ arms and three unique gaps - a small gap $\Delta_3$ and two large gaps $\Delta_2 < \Delta_1 = \Delta_{\max} = 0.4$. The hardness parameter is changed by increasing $\Delta_2$ (from $0.35$ to $0.39$) and bringing it closer to $\Delta_1$. The rewards are normally distributed with $\sigma = 0.05$. We see

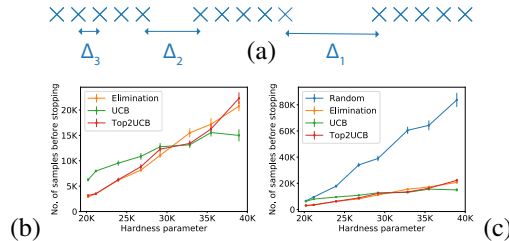

Figure 6: Stopping time experiments.

a linear relationship in Fig. 6(b) which suggests that the sample complexity expression in Theorem 2 is correct up to constants. In Fig. 6(c) we include random sampling and see that our adaptive algorithms require up to 5x fewer samples when run until completion. Fig. 6(c) also shows that our adaptive algorithms always outperform random sampling, and the gains increase with hardness. We used a lower bound based stopping condition for Random, Elimination, Top2UCB, and set $c = 5$ in the UCB stopping condition (value of $c$ chosen empirically as in [13]).

### 7.2 Simulated Data

In the second experiment, we study the performance on a simulated set of means containing two large gaps. The mean distribution plotted in Fig. 7(a) has $K = 24$ arms ($\mathcal{N}(\cdot, 1)$), with two large mean

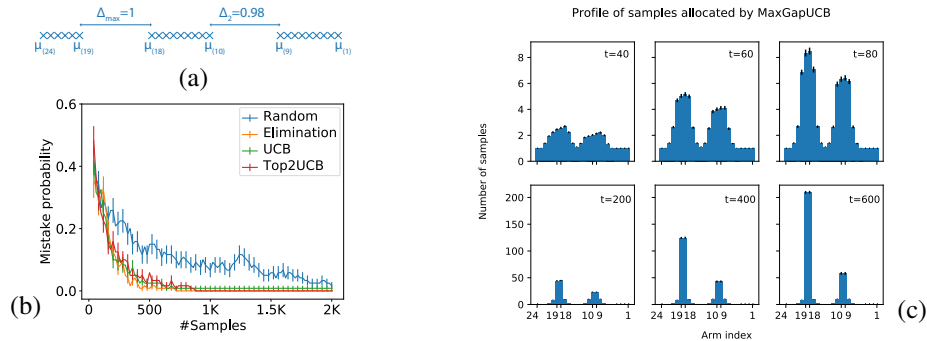

(a)

(b)

(c)

Figure 7: (a) Two large gaps. (b) Clustering error probability for means shown in Fig. 7(a). (c) The profile of samples allocated by `MaxGapUCB` to each arm in (a) at different time steps.

gaps $\Delta_{10,9} = 0.98, \Delta_{19,18} = 1.0$, and remaining small gaps ($\Delta_{i+1,i} = 0.2$ for $i \notin \{9, 18\}$). We expect to see a big advantage for adaptive sampling in this example because almost every sub-optimal arm has a *helper* arm (see Section 5) which can help eliminate it quickly, and adaptive algorithms can then focus on distinguishing the two large gaps. A non-adaptive algorithm on the other hand would continue sampling all arms. We plot the fraction of times $C_1 \neq \{1, \ldots, 18\}$ in 120 runs in Fig. 7(b), and see that the active algorithms identify the largest gap in 8x fewer samples. To visualize the adaptive allocation of samples to the arms, we plot in Fig. 7(c) the number of samples queried for each arm at different time steps by `MaxGapUCB`. Initially, `MaxGapUCB` allocates samples uniformly over all the arms. After a few time steps, we see a bi-modal profile in the number of samples. Since all arms that achieve the largest `U∆` are sampled, we see that several arms that are near the pairs $(10, 9)$ and $(19, 18)$ are also sampled frequently. As time progresses, only the pairs $(10, 9)$ and $(19, 18)$ get sampled, and eventually more samples are allocated to the larger gap $(19, 18)$ among the two.

## 7.3 Streetview Dataset

For our third experiment we study performance on the Streetview dataset [17, 18] whose means are plotted in Fig. 8(a). We have $K = 90$ arms, where each arm is a normal distribution with mean equal to the Borda safety score of the image and standard deviation $\sigma = 0.05$. The largest gap of 0.029 is between arms 2 and 3, and the second largest gap is 0.024. In Fig. 8(b), we plot the fraction of times $\hat{C}_1 \neq \{1, 2\}$ in 120 runs as a function of the number of samples, for four algorithms, viz., random (non-adaptive) sampling, `MaxGapElim`, `MaxGapUCB`, and `MaxGapTop2UCB`. The error bars denote standard deviation over the runs. `MaxGapUCB` and `MaxGapTop2UCB` require 6-7x fewer samples than random sampling.

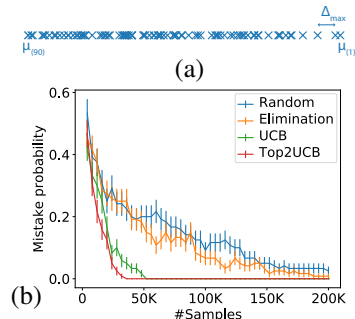

(a)

(b)

Figure 8: (a) Borda safety scores for Streetview images. (b) Probability of returning a wrong cluster.

## 8 Conclusion

In this paper, we proposed the MaxGap-bandit problem: a novel maximum-gap identification problem that can be used as a basic primitive for clustering and approximate ranking. Our analysis shows a novel hardness parameter for the problem, and our experiments show 6-8x gains compared to non-adaptive algorithms. We use simple Hoeffding based confidence intervals in our analysis for simplicity, but better bounds can be obtained using tighter confidence intervals [13]. Several extensions of this basic problem are possible. An $\epsilon$-relaxation of the MaxGap Bandit is useful when the largest and second-largest gaps are close to each other. Other possibilities include identifying the largest gap within a top quantile of the arms, or clustering with a constraint that the returned clusters are of similar cardinality. All of these extensions will likely require new ideas, as it is unclear how to obtain a lower bound for the gap associated with every arm. Finding an instance-dependent lower bound for MaxGap-bandit is an intriguing problem. Finally, one way to cluster the distributions into more than two clusters is to apply the max-gap identification algorithms recursively; however it would be interesting to come up with algorithms that can perform this clustering directly.

## Acknowledgments

Ardhendu Tripathy would like to thank Ervin Tánczos for helpful discussions. The authors would also like to thank the reviewers for their comments and suggestions. This work was partially supported by AFOSR/AFRL grants FA8750-17-2-0262 and FA9550-18-1-0166.

## Footnotes

*Authors contributed equally and are listed alphabetically.

[1]First find the smallest and largest numbers, say $a$ and $b$ respectively. Divide the interval $[a, b]$ into $K + 1$ equal-width bins and map each number to its corresponding bin, while maintaining the smallest and largest number in each bin. Since at least one bin is empty by the pigeonhole principle, the largest gap is between two numbers belonging to different bins. Calculate all gaps between bins and cluster based on the largest of those.

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
