[Supplementary Material]

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

# A  Details for Section 1.2: Comparison to a Naive Algorithm

The naive algorithm first sorts the arms to determine the adjacent arms for every arm, and then runs a best-arm identification bandit algorithm on the gaps to identify the largest gap. An unbiased sample of the gap between two arms can be obtained by taking the difference of the samples from the two arms. Here we analyze the sample complexity of the naive algorithm for a general arrangement of the means.

Consider an arm $i \notin \{(m), (m+1)\}$, and let us analyze the number of times arm $i$ is sampled by the naive algorithm. Let $\Delta_{i,j} = \mu_j - \mu_i$. Then $\Delta_i^r = \min_{j:\Delta_{i,j}>0} \Delta_{i,j}$ is the right gap of arm $i$ ($\Delta_i^l$ is defined analogously). In the first step of the naive algorithm, arm $i$ needs to be sampled at least $(\Delta_i^r)^{-2}$ times to determine its right neighbor. Once the right neighbor has been determined, the best-arm identification requires at least $(\Delta_{\max} - \Delta_i^r)^{-2}$ samples to distinguish arm $i$'s right gap from $\Delta_{\max}$. Since samples from the first step can be reused, the minimum number of samples required by the naive algorithm to rule out arm $i$'s right gap is $(\tilde{\gamma}_i^r)^{-2}$ where

$$\tilde{\gamma}_i^r = \min_{j:\Delta_{i,j}>0} \{\Delta_{i,j}, \Delta_{\max} - \Delta_{i,j}\} \tag{13}$$

We can define $(\tilde{\gamma}_i^l)^{-2}$ analogously, and the naive algorithm collects $\Omega(\tilde{\gamma}_i^{-2})$ from arm $i$, where $\tilde{\gamma}_i = \min\{\tilde{\gamma}_i^r, \tilde{\gamma}_i^l\}$.

The hardness parameter of our active algorithms that is analogous to (13) is given by (4), repeated here for convenience

$$\gamma_i^r := \max_{j:\Delta_{i,j}>0} \min \{\Delta_{i,j}, \Delta_{\max} - \Delta_{i,j}\}. \tag{14}$$

Comparing (14) to (13), we see that $\gamma_i^r > \tilde{\gamma}_i^r$.

For the toy problem discussed in Section 1.2, if we assume that $\Delta_{\min} < \Delta_{\max}/2$, we have that $\tilde{\gamma}_i = \Delta_{\min}$, while $\gamma_i = \Delta_{\max}/2 \, \forall \, i \notin \{(m), (m+1)\}$, which results in $(\Delta_{\max}/\Delta_{\min})^2$ order savings in the number of samples.

# B  Details for Section 4: Confidence Bounds for Gaps

We first explain the mixed integer program formulation for obtaining the upper confidence bounds on the mean gaps in Appendix B.1, and then prove the validity of Algorithm 4 in Appendix B.2.

## B.1  MIP Formulation of Confidence Bounds for Gaps

Conceptually, the confidence intervals on the arm means can be used to construct upper confidence bounds on the mean gaps $\{\Delta_i\}_{i \in [K]}$ in the following manner. Consider all possible configurations of the arm means that satisfy the confidence interval constraints in (5). Each configuration fixes the gaps associated with any arm $a \in [K]$. Then the maximum gap value over all configurations is the upper confidence bound on arm $a$'s gap; we denote it as $\mathtt{U}\Delta_a$.

If we focus on the right gap of arm $a$, the above procedure is equivalent to solving the following optimization problem.

$$\mathtt{U}\Delta_a^r(t) \triangleq \max_{b \in [K] \setminus \{a\}} \max_{\mu_1', \ldots, \mu_K'} \mu_b' - \mu_a' \tag{15}$$

$$\text{subject to: } l_i(t) \leq \mu_i' \leq r_i(t) \quad \forall i \in [K], \text{ and} \tag{16}$$

$$\mu_i' \notin (\mu_a', \mu_b') \, \forall \, i \in [K] \setminus \{a, b\}. \tag{17}$$

Constraint (16) ensures that $\mu_i'$ is in the confidence interval for the mean of arm $i$ at time $t$, and constraint (17) ensures that arm $b$ is the right neighbor of arm $a$.

The constraint (17) is a sorting constraint that can only be formulated using a binary variable. For an arm $i \in [K] \setminus \{a, b\}$ (17) can be formulated using a constant $M$ as

$$\mu_i' \leq \mu_a' + M(1 - z_i), \tag{18a}$$

$$\mu_i' \geq \mu_b' - M z_i, \tag{18b}$$

$$z_i \in \{0, 1\}. \tag{18c}$$

The value of $M$ is chosen to be large number. Replacing constraint (17) by constraints (18a), (18b), (18c) for all $i \in [K] \setminus \{a, b\}$ gives an equivalent optimization problem whose optimum value is $\mathtt{U}\Delta_a^r(t)$. This can be seen to be true by considering the cases based on the value of $z_i$. If $z_i = 0$, $\mu_i' \geq \mu_b'$ and if $z_i = 1$, $\mu_i' \leq \mu_a'$. Because $M$ is chosen to be a large number, in either case $\mu_i' \notin (\mu_a', \mu_b')$ and constraint (17) is satisfied. If constraint (17) is satisfied, then a similar argument allows us to choose the value of $z_i$ that satisfies constraints (18a) and (18b).

## B.2 Validity of Algorithm 4

In this section we find the value of $\mathtt{U}\Delta_a^r(t)$ as defined in (15) by first obtaining an upper bound to it. The proof of the upper bound is *constructive* in nature, showing that the upper bound is actually achievable. That is, (a) there is a set of real numbers $\{\mu_i' : i \in [K]\}$ which satisfy (16), (b) an index $a_*$ which satisfies (17) with $x = \mu_{a_*}'$, such that $\mathtt{U}\Delta_a^r(t) = \mu_{a_*}' - \mu_a'$.

We first find an upper bound to the right gap of an arm $a$ assuming we know its true mean $\mu_a$, but only have confidence intervals for the means of the other arms $\mu_i \in [l_i(t), r_i(t)] \forall i \neq a$.

**Lemma 2.** *If all the arm means are known, the right gap associated with an arm $a \in [K]$ is $\min_{i:\mu_i > \mu_a} \mu_i - \mu_a$; if the domain is empty we say that arm $a$'s right gap is $0$. For any $x \in \mathbb{R}$, define a function $G_a^r(\cdot)$ of the confidence intervals as follows.*

$$G_a^r(x, t) \triangleq \begin{cases} \min_{j:l_j(t) > x} r_j(t) - x & \text{if } \{j : l_j(t) > x\} \neq \emptyset, \\ \max_{j \neq a} r_j(t) - x & \text{otherwise.} \end{cases}$$

*Suppose we know the value of arm $a$'s mean, i.e. $\mu_a$ and the confidence intervals $[l_i(t), r_i(t)] \forall i \neq a$. Then the largest possible right gap of arm $a$ is $G_a^r(\mu_a, t)$.*

*Proof.* We suppose that the right gap of arm $a$ is greater than the upper bound and show a contradiction to the good event (6).

**Case I:** $\{j : l_j(t) > \mu_a\} \neq \emptyset$. Identify the arm $j_* = \arg\min_{j:l_j(t) > \mu_a} r_j(t)$ such that $G_a^r(\mu_a, t) = r_{j_*}(t) - \mu_a$. Let the true right gap for arm $a$ be $\mu_k - \mu_a$. If $k = j_*$, then $\mu_k - \mu_a > r_{j_*}(t) - \mu_a$ would mean that $\mu_{j_*} > r_{j_*}(t)$, which is a contradiction. If $k \neq j_*$ and the right gap is $\mu_k - \mu_a$, then all arms $j \in [K]$ are such that $\mu_j \notin (\mu_a, \mu_k)$. But if $\mu_k - \mu_a > G_a^r(\mu_a, t)$ then $\mu_k > r_{j_*}(t)$, and from the domain in the definition of $j_*$, its left bound $l_{j_*}(t) > \mu_a$. Hence the confidence interval of $j_*$ satisfies $\mu_a < l_{j_*}(t) < r_{j_*}(t) < \mu_k$. If $\mu_{j_*} \notin (\mu_a, \mu_k)$ then $\mu_{j_*} \notin [l_{j_*}(t), r_{j_*}(t)]$ and that is a contradiction.

**Case II:** $\{j : l_j(t) > \mu_a\} = \emptyset$. Identify the arm $j_* = \arg\max_{j \neq a} r_j(t)$ such that $G_a^r(\mu_a, t) = r_{j_*}(t) - \mu_a$. Let the true right gap for arm $a$ be $\mu_k - \mu_a$. If $\mu_k - \mu_a > G_a^r(\mu_a, t)$ then $\mu_k > \max_{j \neq a} r_j(t)$ and that is a contradiction.

Thus the right gap of arm $a$ is at most $G_a^r(\mu_a, t)$. We can achieve this upper bound by choosing the set of means $\{\mu_i' : i \in [K] \setminus a\}$ in the following manner. If the value of $G_a^r(\mu_a, t)$ is given by the first branch, set $\mu_i' = r_i(t) \forall i : r_i(t) > \mu_a$ and $\mu_i' = l_i(t) \forall i : l_i(t) < \mu_a$. Otherwise if the value is given by the second branch set $\mu_{a_*}' = r_{a_*}(t)$ for the arm $a_* \neq a$ which has the largest right bound, and set all other $\mu_i' = l_i(t)$ (*c.f.* Fig. 3 in Section 4). $\qquad\square$

The *left gap* analog of the above proposition can also be proved in a similar manner as above.

**Lemma 3.** *For any $x \in \mathbb{R}$ and arm $a \in [K]$, define a function $G_a^l(\cdot)$ of the confidence intervals as follows.*

$$G_a^l(x, t) \triangleq \begin{cases} x - \max_{j:r_j(t) < x} l_j(t) & \text{if } \{j : r_j(t) < x\} \neq \emptyset, \\ x - \min_{j \neq a} l_j(t) & \text{otherwise.} \end{cases} \tag{19}$$

*Suppose we know $\mu_a$. Using the confidence intervals $[l_i(t), r_i(t)] \forall i \neq a$, an upper bound to the left gap of arm $a$ is $G_a^l(\mu_a, t)$.*

We now replace our knowledge of the true mean value $\mu_a$ by the good event fact that $\mu_a \in [l_a(t), r_a(t)]$ at all times $t$. The following lemma is instrumental in arriving at an upper bound for the right gap of arm $a$ that is consistent with the all the arms' confidence intervals.

**Lemma 4.** *At time $t$, for any arm $a$ its true mean $\mu_a \in [l_a(t), r_a(t)]$ in the good event. Define a subset of arms $\mathcal{I}_a^R(t) \triangleq \{i : l_i(t) \in [l_a(t), r_a(t)]\}$ whose left bounds lie within the confidence interval of arm $a$. Consider a set of $K$ real numbers $\mathcal{P}' \triangleq \{\mu_i' \in [l_i(t), r_i(t)] : i \in [K]\}$, each associated with a corresponding arm. The largest value for the right gap of arm $a$ if the means are $\mathcal{P}'$, i.e.,*

$$\max\{\mu_i' - \mu_a' : \mu_i' > \mu_a', \nexists \mu_j' \in (\mu_a', \mu_i'), i, j \in [K] \setminus a\}$$

*occurs when $\mu_a' = l_i(t)$ for some $i \in \mathcal{I}_a^R(t)$.*

*Proof.* Suppose the largest right gap occurs when $\mu_a' \neq l_i(t)$ for any $i \in \mathcal{I}_a^R(t)$. Note that $a \in \mathcal{I}_a^R(t)$ and hence the set is not empty. We show that the right gap can be larger while still satisfying event (6). Let $l_{i_a}(t) = \max_{i \in \mathcal{I}_a^R(t)}\{l_i(t) < \mu_a'\}$. Collect all arms in the set $\mathcal{J}_a = \{j : \mu_j' \in [l_{i_a}(t), \mu_a']\}$. Consider an alternate bandit model whose arm means are denoted by $\mathcal{Q} \triangleq \{q_i : i \in [K]\}$. We assign

$$q_i = l_{i_a}(t) \,\forall i \in \mathcal{J}_a \text{ and } q_i = \mu_i' \,\forall i \notin \mathcal{J}_a.$$

This mean assignment satisfies $q_i \in [l_i(t), r_i(t)] \,\forall i \in [K]$. This is because by definition of arm $i_a$ in the original bandit model $\mathcal{P}'$, for all arms $j \in \mathcal{J}_a$ their left bounds satisfy $l_j(t) \leq l_{i_a}(t)$. Thus both the original $\mathcal{P}'$ and the alternate $\mathcal{Q}$ are possible bandit models in the good event (6) up till current time $t$. However, the right gap for $a$ is larger in the alternate model $\mathcal{Q}$ as shown next. Let arm $i$ result in the right gap for $a$ in the original model $\mathcal{P}'$, i.e., the right gap

$$\mu_i' - \mu_a', \text{ and } \nexists \mu_j' \in (\mu_a', \mu_i').$$

Then in the alternate model, $q_i = \mu_i', q_a = l_{i_a}(t)$ and there is no mean $q_j \in (l_{i_a}(t), \mu_i')$. Then the right gap of arm $a$ is $\mu_i' - l_{i_a}(t) > \mu_i' - \mu_a'$. This contradicts the supposition that the right gap is the largest possible in the original bandit model $\mathcal{P}'$. $\square$

An analogous lemma for the *left gap* states that for any set of possible arm means $\mathcal{P}'$ that are consistent with the current confidence intervals, the largest possible left gap of arm $a$ occurs when $\mu_a' = r_i(t)$ for some arm $i \in \mathcal{I}_a^L(t) \triangleq \{i : r_i(t) \in [l_a(t), r_a(t)]\}$. Using the above, we can state the upper bound for the gap of an arm $a$ in terms of all the confidence intervals as follows.

**Theorem 3.** *At any time $t$, denote the upper bound to the right (resp. left) gap of arm $a$ by $\mathtt{U}\Delta_a^r(t)$ (resp. $\mathtt{U}\Delta_a^l(t)$). The expressions for these upper bounds in terms of the confidence intervals and the functions $G_a^r(\cdot), G_a^l(\cdot)$ in Lemma 2, Lemma 3 are as follows.*

$$\mathtt{U}\Delta_a^r(t) \triangleq \max\{G_a^r(l_j(t), t) : l_j(t) \in [l_a(t), r_a(t)]\},$$
$$\mathtt{U}\Delta_a^l(t) \triangleq \max\{G_a^l(r_j(t), t) : r_j(t) \in [l_a(t), r_a(t)]\}. \tag{20}$$

*Then an upper bound to the gap associated with arm $a$ at time $t$ is $\max\{\mathtt{U}\Delta_a^r(t), \mathtt{U}\Delta_a^l(t)\}$. Algorithm 4 gives pseudocode that evaluates $\mathtt{U}\Delta_a^r(t)$.*

*Proof.* We argue for the right gap, an analogous proof gives the statement for the left gap. At any time $t$ in the good event $\mu_i \in [l_i(t), r_i(t)] \forall i \in [K]$, in particular any number in the range $[l_a(t), r_a(t)]$ can be potentially the mean of arm $a$. From Lemma 4, we know that for a set of numbers $\mathcal{P}'$ that satisfy all current confidence intervals and also maximize the right gap for arm $a$, the value $\mu_a' = l_i(t)$ for some left bound $l_i(t) \in [l_a(t), r_a(t)]$. If $\mu_a' = l_i(t)$ then by Lemma 2 $G_a^r(l_i(t), t)$ is the largest possible value for arm $a$ in the bandit model $\mathcal{P}'$. Taking the maximum over all arms in the set $\mathcal{I}_a^R(t) = \{i \in [K] : l_i(t) \in [l_a(t), r_a(t)]\}$, we get the right gap upper bound $\mathtt{U}\Delta_a^r(t)$.

We note that the value $\mathtt{U}\Delta_a^r(t)$ is achievable by an assignment of means that satisfy the confidence bounds at time $t$. Without loss of generality, assume $\mathtt{U}\Delta_a(t) = \mathtt{U}\Delta_a^r(t) = G_a^r(l_{a_*}(t), t)$ for some arm $a_*$. One can assign $\mu_a = l_{a_*}(t)$ and other means in a way similar to that in the proof of Lemma 2 to obtain a right gap for arm $a$ equal to the value $G_a^r(l_{a_*}(t), t)$. $\square$

## C  Details for Section 5: Accuracy

**Theorem 1.** *With probability $1 - \delta$, `MaxGapElim`, `MaxGapUCB` and `MaxGapTop2UCB` cluster the arms according to the maximum gap, i.e., they satisfy (3).*

*Proof.* Recall that the true maximum gap exists between arms $(m)$ and $(m+1)$. The algorithms return a wrong clustering $\texttt{U}\Delta_{(m)}(t) < \texttt{L}\Delta(t)$ for any time $t$. We show that this leads to a contradiction if the good event (6) holds.

Assume (6) holds and $\texttt{U}\Delta_{(m)}(t) < \texttt{L}\Delta(t)$ at some time $t$. Recall that $\texttt{L}\Delta(t)$ is computed using (9), and let $(s)_t$ be the maximizer in (9). Let $a$ be such that $a \in \{(1)_t, \ldots, (s)_t\}$ and $a+1 \in \{(s+1)_t, \ldots, (K)_t\}$. If (3) holds, we have that

$$\Delta_{\max} \leq \texttt{U}\Delta_{(m)}(t) < \texttt{L}\Delta(t) \overset{(a)}{\leq} l_a(t) - r_{a+1}(t) \leq \mu_a - \mu_{a+1},$$

where (a) holds because $\texttt{L}\Delta(t)$ is the minimum gap between a left confidence interval in $\{(1)_t, \ldots, (s)_t\}$ and a right confidence interval in $\{(s+1)_t, \ldots, (K)_t\}$. This contradicts the fact that $\Delta_{\max}$ is the largest gap. $\qquad\square$

# D  Sample Complexity: Proof of Theorem 2

To state our sample complexity bounds we use a constant $\alpha$ defined as follows [7].

**Remark 1.** *There exists constant $\alpha$ such that for all $x > 0$, if the number of samples $s \geq \alpha \frac{\log(K/\delta x)}{x^2}$, then $c_s \leq x$, where $c_s$ is the confidence interval given by (5).*

## D.1  Sample Complexity of `MaxGapElim`

**Early Stopping Rule for Clustering**: In the pseudocode in Algorithm 1, `MaxGapElim` stops when the size of the active set $|A| \leq 2$ (line 7). However, if we are only interested in clustering the arms according to the maximum gap and not interested in the identities of the arms which share the maximum gap (arms $(m), (m+1)$), we can stop earlier as follows. Assume that (9) is greater than 0 and let $(k_*)_t$ be the maximizer. This partitions the arms into the sets $\{(1)_t, \ldots, (k_*)_t\}$ and $\{(k_*+1)_t, \ldots, (K)_t\}$. `MaxGapElim` can terminate when the maximum left gap of all arms in $\{(1)_t, \ldots, (k_*)_t\}$ and the maximum right gap of all arms in $\{(k_*+1)_t, \ldots, K\}$ are both less than the lower bound $\texttt{L}\Delta(t)$. The termination condition can be expressed as $S = 1$, where

$$S = \mathbb{1}\{\texttt{U}\Delta_a^r(t) < \texttt{L}\Delta(t), \forall a : l_a(t) \geq l_{(k_*)_t}(t\} \cdot \mathbb{1}\{\texttt{U}\Delta_a^l(t) < \texttt{L}\Delta(t), \forall a : r_a(t) \leq l_{(k_*+1)_t}(t)\}. \tag{21}$$

To account for the lower sample complexity as a result of the stopping rule for clustering, we modify (10) and (11) and define new parameters that yield an improved sample complexity than that stated in Theorem 2. Define

$$\rho_a^r = \max\left\{ \max_{j:\Delta_{a,j}>0} \left(\min\{\Delta_{a,j}/4, ((\Delta_{\max} - \Delta_{a,j})/8)\}\right), ((\Delta_{\max} - \Delta_{a,1})/8)\right\}, \tag{22}$$

$$\rho_a^l = \max\left\{ \max_{j:\Delta_{a,j}<0} \left(\min\{\Delta_{a,j}/4, ((\Delta_{\max} - \Delta_{j,a})/8)\}\right), ((\Delta_{\max} - \Delta_{a,K})/8)\right\}, \tag{23}$$

where just like in (10), the maxima assumed to be infinity if there is no $j$ that satisfies the constraint under the inner maximization. We define $\rho_a = \min\{\rho_a^r, \rho_a^l\}$ as before and state our improved sample complexity bound for `MaxGapElim` next.

**Theorem 4.** *With probability at least $1 - \delta$, the sample complexity of `MaxGapElim` is bounded by*

$$H = \alpha \sum_{\substack{a \in [K]: \\ a \notin \{(m),(m+1)\}}} \frac{\log(K/\delta\rho_a)}{\rho_a^2}.$$

*Proof.* Arm $a$ is eliminated in `MaxGapElim` when $\texttt{U}\Delta_a(t) < \texttt{L}\Delta(t)$, whee $\texttt{U}\Delta_a(t)$ is defined as the maximum of the left and right gap upper bounds (see Section 4). Lemma 6 and Lemma 7 prove that the sufficient condition for each of these upper bounds to be les than $\texttt{L}\Delta(t)$ is $c_{T_a(t)} \leq \rho_a$. The result then follows by Remark 1. $\qquad\square$

**Lemma 5.** *If the good event* (6) *holds, then for all $a \in [K]$, for all $t \in \mathbb{N}$,*

$$l_a(t) \geq \mu_a - 2c_{T_a(t)} \text{ and } r_a(t) \leq \mu_a + 2c_{T_a(t)}$$

*where $c_s = \sqrt{\frac{\beta_\delta(s)}{s}}$.*

*Proof.* We have

$$\hat{\mu}_a(t) + c_{T_a(t)} \overset{(a)}{\geq} \mu_a \Rightarrow l_a(t) = \hat{\mu}_a(t) - c_{T_a(t)} \geq \mu_a - 2c_{T_a(t)}.$$

Similarly,

$$\hat{\mu}_a(t) - c_{T_a(t)} \overset{(a)}{\leq} \mu_a \Rightarrow r_a(t) = \hat{\mu}_a(t) + c_{T_a(t)} \leq \mu_a + 2c_{T_a(t)}.$$

In both the equations above, $(a)$ holds by (6). $\qquad\square$

**Lemma 6.** *Assume* (6) *holds, and consider $a \neq m+1$. In* `MaxGapElim` *if $t$ is such that $c_{T_a(t)} \leq \rho_a^r$, then*

$$\text{U}\Delta_a^r(t) < \text{L}\Delta(t).$$

*Proof.* Note that at time $t$ in Algorithm 1, $T_a(t) = t$ and $c_{T_a(t)} = c_t$ for all arms $a \in A$. Assume (6) holds. We have $c_t < \rho_a^r < \Delta_{\max}/4$. This implies that

$$l_m(t) \overset{(a)}{\geq} \mu_m - 2c_t = \mu_{m+1} + \Delta_{\max} - 2c_t \overset{(a)}{\geq} r_{m+1}(t) + \Delta_{\max} - 4c_t \geq r_{m+1}(t). \qquad (24)$$

where (a) holds by Lemma 5.

From (24) we have that

$$\text{L}\Delta(t) \geq l_m(t) - r_{m+1}(t) \geq \Delta_{\max} - 4c_t \qquad (25)$$

Recall from (22) that for $a \neq 1$,

$$\rho_a^r = \max \left\{ \max_{j:\Delta_{a,j}>0} \left( \min\{\Delta_{a,j}/4, ((\Delta_{\max} - \Delta_{a,j})/8)\} \right), ((\Delta_{\max} - \Delta_{a,1})/8) \right\}. \qquad (26)$$

There are two terms in $\rho_a^r$ and $c_t$ could be less than either of these terms. First, suppose that

$$c_t < \max_{j:\Delta_{a,j}>0} \left( \min\{\Delta_{a,j}/4, ((\Delta_{\max} - \Delta_{a,j})/8)\} \right),$$

and let

$$e = \arg\max_{j:\Delta_{a,j}>0} \left( \min\{\Delta_{a,j}/4, ((\Delta_{\max} - \Delta_{a,j})/8)\} \right). \qquad (27)$$

For any arm $j$ such that $\Delta_{\max} < \Delta_{a,j}$, the inner minimum in (27) will be negative. On the other hand, since $a \neq m+1$, there must exist an arm $j$ such that $\Delta_{\max} > \Delta_{a,j}$, and for such an arm $j$ the inner minimum will be positive. Since $e$ is the arm that maximizes the inner minimum, the inner minimum must be positive for $e$. Thus we have that $\Delta_{\max} > \Delta_{a,e}$.

From (26), (27), we have that

$$c_t < \Delta_{a,e}/4 \quad \text{and} \quad c_t < (\Delta_{\max} - \Delta_{a,e})/8. \qquad (28)$$

Since $c_t < \Delta_{a,e}/4$, by following an argument similar to (24) we have that $l_e(t) \geq r_a(t)$, and hence the first branch of (7) will be used to compute $\text{U}\Delta_a^r(t)$. Hence we have

$$\text{U}\Delta_a^r(t) \overset{(a)}{\leq} r_e(t) - l_a(t) \overset{(b)}{\leq} \Delta_{a,e} + 4c_t \overset{(c)}{\leq} \Delta_{\max} - 4c_t \overset{(d)}{\leq} \text{L}\Delta(t)$$

where $(a)$ follows from (7) and (8), $(b)$ holds from Lemma 5, $(c)$ follows by (28), and $(d)$ holds by (25).

For the second case, assume

$$c_t < (\Delta_{\max} - \Delta_{a,1})/8.$$

Let $e = \arg\max_{i \neq a} r_i(t)$. From (8), we have that

$$\text{U}\Delta_a^r(t) \leq r_e(t) - l_a(t) \overset{(a)}{\leq} \Delta_{a,e} + 4c_t \overset{(b)}{\leq} \Delta_{a,1} + 4c_t \overset{(c)}{\leq} \Delta_{\max} - 4c_t \overset{(c)}{\leq} L(t),$$

where (a) holds by Lemma 5, (b) holds by the case assumption, and (c) holds by (25). $\qquad\square$

**Lemma 7.** *Assume* (6) *holds, and consider* $a \neq m$. *In* `MaxGapElim` *if* $t$ *is such that* $c_t \leq \rho_a^l$, *then*

$$\mathtt{U\Delta}_a^l(t) < \mathtt{L\Delta}(t)$$

*Proof.* The proof is analogous to the proof of Lemma 6. $\qquad\qquad\qquad\qquad\qquad\square$

### D.2 Sample Complexity of `MaxGapUCB`

For the sample complexity analysis of `MaxGapUCB` , we use a modified version of the left and right confidence bounds introduced in (5). We redefine

$$l_i'(t) \triangleq \max_{s \leq t} l_i(s), \qquad r_i'(t) \triangleq \min_{s \leq t} r_i(s). \tag{29}$$

The nice property that these bounds have is that $[l_i'(t), r_i'(t)] \subseteq [l_i(s), r_i(s)]$ for all $t \geq s$. Lemma 9 shows that these modified bounds retain the same confidence guarantee for the arm mean values as the original confidence bounds. In what follows, we will exclusively use the modified confidence bounds (except in Lemma 9 where we show they are correct). We drop the prime symbol in their notation for brevity and henceforth $l_i(t), r_i(t)$ denote the modified confidence bounds given in (29).

We state and prove our main sample complexity result in Theorem 5.

**Theorem 5.** *With probability at least* $1 - \delta$, *the number of times* `MaxGapUCB` *samples a sub-optimal arm, i.e. an arm* $i \notin \{(m), (m+1)\}$, *is upper bounded by* $6\alpha\gamma_i^{-2}\log(K/\delta\gamma_i)$. *The constant* $\alpha$ *is defined in Remark 1. Thus, the number of times* `MaxGapUCB` *samples suboptimal arms is*

$$H = 6\alpha \sum_{\substack{i \in [K]: \\ i \notin \{(m),(m+1)\}}} \frac{\log(K/\delta\gamma_i)}{\gamma_i^2}.$$

*Proof.* We show that the result holds true as long as the confidence intervals for the means are correct (6). Let

$$\tau_r = \alpha \frac{\log(K/\delta\gamma_i^r)}{(\gamma_i^r)^2}, \quad \text{and} \quad \tau_l = \alpha \frac{\log(K/\delta\gamma_i^l)}{(\gamma_i^l)^2}, \tag{30}$$

where $\alpha$ is defined in Remark 1. Note that $\mathtt{U\Delta}_{(m)}(t) = \mathtt{U\Delta}_{(m+1)}(t) \geq \Delta_{\max} \forall t$. Arm $i$ is sampled either because $\mathtt{U\Delta}_i^r$ is the largest or $\mathtt{U\Delta}_i^l$ is the largest. We prove in Lemma 8 below that when $i$ is sampled $3\tau_r$ times due to its right gap, $\mathtt{U\Delta}_i^r < \Delta_{\max}$. Hence `MaxGapUCB` will not sample $i$ due to its right gap more than $3\tau_r$ times because beyond this point $\mathtt{U\Delta}_{(m)}$ will be higher. It can similarly be proved that when $i$ is sampled $3\tau_l$ times due to its left gap, $\mathtt{U\Delta}_i^l < \Delta_{\max}$. Thus, arm $i$ will be sampled at most $3(\tau_r + \tau_l) \leq 6\max\{\tau_r, \tau_l\}$ times. $\qquad\qquad\square$

To ease the explanation, we only focus on the right gap of $i$ from here onwards and set

$$\tau = \alpha \frac{\log(K/\delta\gamma_i^r)}{(\gamma_i^r)^2}. \tag{31}$$

Furthermore, in the lemmas below, we only focus on samples of $i$ drawn when $\mathtt{U\Delta}_i^r$ was the largest upper bound. With a slight overload of notation, let $t(i, s)$ denote the (random) smallest time when arm $i$ has been sampled $s$ times by `MaxGapUCB` (owing to its right gap).

**Lemma 8.** *With probability* $1 - \delta$, $\mathtt{U\Delta}_i^r(t(i, 3\tau)) < \Delta_{\max}$.

*Proof.* Since the proof is long and technical, we first give an outline of the entire proof.

*Outline:* Define $i_r^t$ and $i_l^t$ to be the arms that form the left and right boundaries of $\mathtt{U\Delta}_i^r(t)$ (the two arms that result in the maximum value of (8) at $t$). By this definition, $\mathtt{U\Delta}_i^r(t) = G_i^r(l_{i_l^t}(t), t) = r_{i_r^t}(t) - l_{i_l^t}(t)$. Consider the arms used in computing $\mathtt{U\Delta}_i^r(t(i, 3\tau))$, i.e. $i_l^{t(i,3\tau)}, i_r^{t(i,3\tau)}$, and denote them as $i_l, i_r$ for brevity. Fig. 9 shows the confidence intervals of $i_l$ in blue and those of $i_r$ in green. Initially the confidence intervals are large, i.e., the width between the right and left bounds of arm $i$ is greater than $\Delta_{\max}$ before time $t(i, \tau)$. After $t(i, 2\tau)$ rounds of `MaxGapUCB`, the confidence interval of $i$ will have shrunk. However, note that the right gap of arm $i$ involves either $i_l$ and/or $i_r$. Since

Figure 9: Illustration of left and right confidence bounds during a run of `MaxGapUCB` at three different times, the argument $t$ for the bounds are omitted. Arms $i_l, i_r$ are such that $\mathrm{U}\Delta_i^r(t(i,3\tau)) = r_{i_r}(t(i,3\tau)) - l_{i_l}(t(i,3\tau))$.

`MaxGapUCB` samples *all* arms that can attain the highest gap upper bound, it turns out that it will also sample $i_l$ enough times to make $r_{i_l}(t(i,2\tau)) - l_i(t(i,2\tau)) < \Delta_{\max}$. If $i$ is still sampled after $t(i,2\tau)$ rounds due to its right gap, then its gap upper bound must involve an arm which is disjoint from $i$'s confidence interval, such as the arm $i_r$. Then from $t(i,2\tau)$ to $t(i,3\tau)$, `MaxGapUCB` samples $i_l$ and $i_r$ enough times to make $\mathrm{U}\Delta_i^r(t(i,3\tau)) = r_{i_r}(t(i,3\tau)) - l_{i_l}(t(i,3\tau)) < \Delta_{\max}$.

We divide the proof into four parts. In the first part, we divide all the arms into subsets (which we refer to as levels). These subsets are defined such that arms within a subset obey collective properties, that we study in some of the subsequent lemmas. In the second and third part, we prove that arms $i_r^{t(i,3\tau)}$ and $i_l^{t(i,3\tau)}$ are always sampled whenever $i$ is sampled from $[t(i,2\tau), t(i,3\tau)]$. Finally in part four, we use these arms to argue that $\mathrm{U}\Delta_i^r(t(i,3\tau)) < \Delta_{\max}$.

**Level Sets**:
At any time $t$, we can identify three subsets of arms with respect to arm $i$ that we refer to as level 0, level 1, and level 2 arms respectively, and argue that the arms that define $\mathrm{U}\Delta_i^r(t)$ must lie in one of these subsets. These levels sets are defined as follows. Let

$$\mathcal{A}_i^0(t) = \{a \in [K] : l_i(t) \le r_a(t) < r_i(t)\}, \tag{32}$$

$$\mathcal{A}_i^1(t) = \{a \in [K] : l_a(t) \le r_i(t) \le r_a(t)\}, \tag{33}$$

$$\mathcal{A}_i^2(t) = \left\{a \in [K] : r_i(t) < l_a(t) \le \min_{j:l_j(t)>r_i(t)} r_j(t)\right\}. \tag{34}$$

From their definitions the three subsets are pairwise disjoint at every $t \in \mathbb{N}$. Let

$$\mathcal{A}_i(t) = \mathcal{A}_i^0(t) \cup \mathcal{A}_i^1(t) \cup \mathcal{A}_i^2(t) \tag{35}$$

denote the union of the three levels. From the definition of $\mathrm{U}\Delta_i^r(t)$ in (8) and (32), (33), the arm

$$i_l^t \in \mathcal{A}_i^0(t) \cup \mathcal{A}_i^1(t) \,\forall\, t. \tag{36}$$

Lemma 10 proves that the arm $i_r^t \in \mathcal{A}_i(t) \,\forall\, t$. Thus at any time $t$, only arms in $\mathcal{A}_i(t)$ are relevant for the right gap of arm $i$.

Suppose $t(i,3\tau) < \infty$, i.e., arm $i$ is sampled at least $3\tau$ times. To avoid clutter, we let

$$i_r = i_r^{t(i,3\tau)} \quad \text{and} \quad i_l = i_l^{t(i,3\tau)},$$

and use the full notation $i_r^t$ for $t \ne t(i,3\tau)$. We next argue that $i_r$ and $i_l$ must be sampled $\tau$ times before $t(i,3\tau)$.

**$i_r$ must have been sampled at least $\tau$ times before $t(i,3\tau)$**:
By Lemma 10, $i_r \in \mathcal{A}_i(t(i,3\tau))$. From Corollary 2, $i_r \in \mathcal{A}_i(s) \,\forall\, s \in [t(i,2\tau), t(i,3\tau)]$. If $i_r \in \mathcal{A}_i^0(s) \cup \mathcal{A}_i^1(s)$ for any $s \in [t(i,2\tau), t(i,3\tau)]$, then $r_{i_r}(t(i,3\tau)) - l_i(t(i,3\tau)) \le r_{i_r}(s) - l_i(s) < \Delta_{\max}$ by Lemma 9 and Lemma 11, and we are done. Let us hence look at the case when $i_r \in \mathcal{A}_i^2(s) \,\forall\, s \in [t(i,2\tau), t(i,3\tau)]$. We have by Lemma 11 that $i_r^s \in \mathcal{A}_i^2(s) \,\forall\, s \in [t(i,2\tau), t(i,3\tau)]$, and Lemma 13 then implies that $i_r$ must be sampled whenever $i$ was sampled for $s \in [t(i,2\tau), t(i,3\tau)]$. Hence $i_r$ is sampled at least $\tau$ times before $t(i,3\tau)$.

**$i_l$ must have been sampled at least $\tau$ times before $t(i, 3\tau)$:**
From Corollary 2, since the level of an arm cannot decrease from 2 to 1, $i_l \in \mathcal{A}_i^0(s) \cup \mathcal{A}_i^1(s)$ for all $s \in [t(i, 2\tau), t(i, 3\tau)]$. If $i_l \in \mathcal{A}_i^1(s) \, \forall \, s \in [t(i, 2\tau), t(i, 3\tau)]$, then $i_l$ is sampled $\tau$ times whenever $i$ is sampled by Lemma 13.
On the other hand, if $i_l \in \mathcal{A}_i^0(s)$ for some $s \in [t(i, 2\tau), t(i, 3\tau)]$, let $\mathtt{U\Delta}_i^r(s) = r_{i_r^s}(s) - l_{i_l^s}(s)$. We consider two cases, $r_{i_l}(s) < l_{i_l^s}(s)$ and $r_{i_l}(s) \geq l_{i_l^s}(s)$. First, if $r_{i_l}(s) < l_{i_l^s}(s)$, then $l_{i_l}(s') < r_{i_l}(s') < l_{i_l^s}(s')$ for all $s' \geq s$ by Lemma 9. Since $i_l^s \in \mathcal{A}_i^0(s) \cup \mathcal{A}_i^1(s)$, we have by Lemma 9 and Lemma 11 that

$$r_{i_l^s}(t(i, 3\tau)) - l_{i_l}(t(i, 3\tau)) \leq r_{i_l^s}(t(i, 3\tau)) - l_i(t(i, 3\tau)) \leq \Delta_{\max}. \tag{37}$$

By the definition of $\mathtt{U\Delta}_i^r$ in (8) we have that

$$\mathtt{U\Delta}_i^r(t(i, 3\tau)) = G_i^r(l_{i_l}(t(i, 3\tau)), t(i, 3\tau)) \leq r_{i_l^s}(t(i, 3\tau)) - l_{i_l}(t(i, 3\tau)). \tag{38}$$

(38) and (37) imply that $\mathtt{U\Delta}_i^r(t(i, 3\tau)) \leq \Delta_{\max}$, and we are done. For the second case, suppose $r_{i_l}(s) \geq l_{i_l^s}(s)$. Lemma 12 then gives that arm $i_l$ is also sampled at time $s$. Thus, we have shown that either $\mathtt{U\Delta}_i^r(t(i, 3\tau)) < \Delta_{\max}$, or $i_l$ is sampled whenever $i$ is sampled in $[t(i, 2\tau), t(i, 3\tau)]$.

We now show that $\mathtt{U\Delta}_i^r(t(i, 3\tau)) < \Delta_{\max}$.

**$\mathtt{U\Delta}_i^r(t(i, 3\tau)) < \Delta_{\max}$:**
Recall that $T_i(t(i, 3\tau)), T_{i_l}(t(i, 3\tau)), T_{i_r}(t(i, 3\tau))$ are all larger than $\tau$. Let

$$j_* = \underset{j: 0 < \Delta_{i,j} < \Delta_{\max}}{\arg\max} \min\{\Delta_{i,j}/4, (\Delta_{\max} - \Delta_{i,j})/4\}$$

be the maximizer in (10), and note that $\mu_i < \mu_{j_*}$ by definition. Also note that $\tau$ and $\gamma_i^r$ are defined in (31) and (10) respectively so that

$$4c_\tau \leq \Delta_{\max} - \Delta_{i,j_*} \quad \text{and} \quad 4c_\tau \leq \Delta_{i,j_*} \tag{39}$$

We split the proof into various cases depending on the ordering of the means $\mu_i, \mu_{i_l}, \mu_{i_r}, \mu_{j_*}$. First, note that if $\mu_{i_r} \leq \mu_{i_l}$, then

$$\mathtt{U\Delta}_i^r(t(i, 3\tau)) = r_{i_r}(t(i, 3\tau)) - l_{i_l}(t(i, 3\tau)) \leq \mu_{i_r} - \mu_{i_l} + 4c_\tau \leq \Delta_{\max}$$

by (39). Second, if $\max\{\mu_i, \mu_{i_l}\} < \mu_{i_r} < \mu_{j_*}$, then

$$\mathtt{U\Delta}_i^r(t(i, 3\tau)) \leq r_{i_r}(t(i, 3\tau)) - l_{i_l}(t(i, 3\tau)) \leq r_{i_r}(t(i, 3\tau)) - l_i(t(i, 3\tau))$$
$$\leq \mu_{i_r} - \mu_i + 4c_\tau \leq \Delta_{i,j_*} + 4c_\tau \leq \Delta_{\max}$$

by (39). Third, we show that it cannot be the case that $\mu_i < \mu_{j_*} < \mu_{i_l} < \mu_{i_r}$. Assume to the contrary. This implies that

$$l_{i_l}(t(i, 3\tau)) - r_i(t(i, 3\tau)) \geq \mu_{i_l} - \mu_i - 4c_\tau \geq \mu_{j_*} - \mu_i - 4c_\tau > 0,$$

which contradicts Eq. (36). Fourth, it cannot be the case that $\mu_{i_l} < \mu_{i_r} < \mu_i < \mu_{j_*}$, because $i_r \in \mathcal{A}_i^2(t(i, 3\tau))$ by Lemma 11. The only case that remains is $\max\{\mu_i, \mu_{i_l}\} < \mu_{j_*} < \mu_{i_r}$, which we prove next by showing that $T_{j_*}(t(i, 3\tau)) \geq \tau$.

**$\underline{\max\{\mu_i, \mu_{i_l}\} < \mu_{j_*^r} < \mu_{i_r}}$:**

For any time $s \in [t(i, 2\tau), t(i, 3\tau)]$ such that $j_* \in \mathcal{A}_i^1(s) \cup \mathcal{A}_i^2(s)$, we have by Lemma 11 and Lemma 13 that $j_*$ is sampled whenever $i$ is sampled. Thus we only need to focus on times $s$ when $j_* \in \mathcal{A}_i^0(s)$.

Suppose now that $j_* \in \mathcal{A}_i^0(s)$ for some $s \in [t(i, 2\tau), t(i, 3\tau)]$ when $i$ was sampled and $\mathtt{U\Delta}_i^r(s) = r_{i_r^s}(s) - l_{i_l^s}(s)$. Recall that $i_l^s \in \mathcal{A}_i^0(s) \cup \mathcal{A}_i^1(s)$. We consider two cases depending on whether $l_{i_l^s}(t(i, 3\tau)) > l_{i_l}(t(i, 3\tau))$ or $l_{i_l^s}(t(i, 3\tau)) \leq l_{i_l}(t(i, 3\tau))$.

- $l_{i_l^s}(t(i, 3\tau)) > l_{i_l}(t(i, 3\tau))$: We have

$$\mathtt{U\Delta}_i^r(t(i, 3\tau)) = G(l_{i_l}(t(i, 3\tau)), t(i, 3\tau)) \overset{(a)}{\leq} r_{i_l^s}(t(i, 3\tau)) - l_{i_l}(t(i, 3\tau))$$
$$\leq r_{i_l^s}(t(i, 3\tau)) - l_i(t(i, 3\tau)) \overset{(b)}{\leq} r_{i_l^s}(s) - l_i(s) \overset{(c)}{\leq} \Delta_{\max},$$

where $(a)$ holds by (7), $(b)$ holds by Lemma 9, and $(c)$ holds by Lemma 11.

- $l_{i_l^s}(t(i,3\tau)) \leq l_{i_l}(t(i,3\tau))$: Since $\max\{\mu_i, \mu_{i_l}\} < \mu_{j_*}$, we have $l_{i_l}(t) \leq r_{j_*}(t) \,\forall\, t$. Hence, $l_{i_l^s}(t(i,3\tau)) < r_{j_*}(t(i,3\tau))$, and Lemma 9 implies that $l_{i_l^s}(s) \leq r_{j_*}(s)$. Recall that $s$ is a time such that $j_* \in \mathcal{A}_i^0(s)$, and hence
$$r_{j_*}(s) - l_{i_l^s}(s) \leq r_{j_*}(s) - l_i(s) \leq \Delta_{\max}.$$

  Now, since $i$ is sampled at time $s$, we have $\mathtt{U}\Delta_i^r(s) > \Delta_{\max}$, and (7) then implies that $l_{j_*}(s) < l_{i_l^s}(s)$. Hence by Lemma 12 $\mathtt{MaxGapUCB}$ must also sample arm $j_*$ at time $s$.

This proves that $T_{j_*^r}(t(i,3\tau)) \geq \tau$. We use this to prove that $\mathtt{U}\Delta_i^r(t(i,3\tau)) < \Delta_{\max}$ as follows. First note that
$$l_{j_*^r}(t(i,3\tau)) - r_i(t(i,3\tau)) \geq p_{j_*^r} - p_i - 4c_\tau \geq 0.$$

Second, since arm $i_l \in \mathcal{A}_i^0(t(i,3\tau)) \cup \mathcal{A}_i^1(t(i,3\tau))$, and hence
$$l_{i_l}(t(i,3\tau)) < r_i(t(i,3\tau)) \leq l_{j_*}(t(i,3\tau)).$$

Hence
$$\mathtt{U}\Delta_i^r(t(i,3\tau)) = G_i^r(l_{i_l}(t(i,3\tau)), t(i,3\tau)) \leq r_{j_*^r}(t(i,3\tau)) - l_{i_l}(t(i,3\tau))$$
$$\leq r_{j_*^r}(t(i,3\tau)) - l_i(t(i,3\tau)) \leq \mu_{j_*^r} - \mu_i + 4c_\tau \leq \Delta_{\max}.$$
$\square$

**Lemma 9.** *Over the sigma-algebra generated by all the arm rewards up till any time $t \in \mathbb{N}$, we have that*
$$\mathbb{P}\left(\forall t \in \mathbb{N}, \forall i \in [K], \mu_i \in \left[\max_{t' \leq t} l_i(t'), \min_{t' \leq t} r_i(t')\right]\right) = \mathbb{P}(\forall t \in \mathbb{N}, \forall i \in [K], \mu_i \in [l_i(t), r_i(t)]). \tag{40}$$

*Proof.* Let $E'$ be the event in the LHS of (40) and let $E$ be the good event. First we show that $E' \subseteq E$. The event $E'$ implies that at any time $t$ and for any arm $i$, we have that
$$\mu_i \in \left[\max_{t' \leq t} l_i(t'), \min_{t' \leq t} r_i(t')\right] \implies \mu_i \in [l_i(t'), r_i(t')] \,\forall t' \leq t.$$

Hence the good event is true in this case.

Now we show that $E \subseteq E'$. Suppose that $E'$ is not true, so there is a time $t$ and arm $i$ such that $\mu_i \notin [\max_{t' \leq t} l_i(t'), \min_{t' \leq t} r_i(t')]$. Choose two time instants $s_l, s_r \in \mathbb{N}$ such that $s_l \in \arg\max_{t' < t} l_i(t'), s_r \in \arg\min_{t' < t} r_i(t')$. Then the supposition implies that either
$$\mu_i \notin [l_i(s_l), r_i(s_l)] \quad \text{or/and} \quad \mu_i \notin [l_i(s_r), r_i(s_r)].$$

Either of the above statements imply that the good event is not true. Hence $E \implies E'$. $\square$

**Corollary 1.** *For any two time instants $s, t \in \mathbb{N}$ if $s < t$ then $\mathtt{U}\Delta_i^r(s) \geq \mathtt{U}\Delta_i^r(t)$.*

*Proof.* The quantity $\mathtt{U}\Delta_i^r(t)$ is defined in (15) as an optimization problem over a set of $K$ real numbers $\mathcal{P}' = \{\mu_i' \in [l_i(t), r_i(t)] : i \in [K]\}$. For a time $s < t$, the $\mathtt{U}\Delta_i^r(s)$ is an optimization over $\mathcal{P}'' = \{\mu_i'' \in [l_i(s), r_i(s)] : i \in [K]\}$. Lemma 9 states that $[l_i(t), r_i(t)] \subseteq [l_i(s), r_i(s)]$, hence we have that $\mathtt{U}\Delta_i^r(s) \geq \mathtt{U}\Delta_i^r(t)$. $\square$

**Corollary 2.** *For all $k \in [K]$ if $k \in \mathcal{A}_i^2(t)$ then $k \in \mathcal{A}_i(s)$ at all time instants $s \leq t$. If $k \in \mathcal{A}_i^2(t)$ then $k \notin \mathcal{A}_i^0(s') \cup \mathcal{A}_i^1(s')$ at all $s' \geq t$.*

*Proof.* Define $\mathcal{J}(t) \triangleq \{j \in [K] : l_j(t) > r_i(t)\}$. For any $s \leq t$, if $j \in \mathcal{J}(s)$ then using Lemma 9,
$$l_j(t) \geq l_j(s) > r_i(s) \geq r_i(t) \implies j \in \mathcal{J}(t). \tag{41}$$

Hence if $k \in \mathcal{A}_i^2(t)$, from (34) we have that $l_k(t) \leq \min_{j \in \mathcal{J}(t)} r_j(t)$ and we get
$$l_k(s) \leq l_k(t) \leq \min_{j \in \mathcal{J}(t)} r_j(t) \overset{(a)}{\leq} \min_{j \in \mathcal{J}(t)} r_j(s) \overset{(b)}{\leq} \min_{j \in \mathcal{J}(s)} r_j(s),$$

where the inequality (a) is true because of Lemma 9 and inequality (b) is true as $\mathcal{J}(s) \subseteq \mathcal{J}(t)$ by (41). This implies that $k \in \mathcal{A}_i^0(s) \cup \mathcal{A}_i^1(s) \cup \mathcal{A}_i^2(s) = \mathcal{A}_i(s)$.

If $k \in \mathcal{A}_i^2(t)$ we have that $r_i(t) < l_k(t)$. At any $s' \geq t$, from Lemma 9 we have that $r_i(s') \leq r_i(t) < l_k(t) \leq l_k(s')$, i.e., the arm $k \notin \mathcal{A}_i^0(s') \cup \mathcal{A}_i^1(s')$. $\qquad \square$

**Lemma 10.** *The arms $i_r^t, i_l^t$ are such that $\mathtt{U}\Delta_i^r(t) = r_{i_r^t}(t) - l_{i_l^t}(t)$. For the sets as defined in (32), (33), (34) the arm $i_r^t \in \mathcal{A}_i(t) \triangleq \mathcal{A}_i^0(t) \cup \mathcal{A}_i^1(t) \cup \mathcal{A}_i^2(t)$.*

*Proof.* Suppose arm $i_r^t \notin \mathcal{A}_i(t)$, then either $r_{i_r^t}(t) < l_i(t)$ which would give a negative value for $\mathtt{U}\Delta_i^r(t)$, or we have that $l_{i_r}(t) > \min_{a:l_a(t)>r_i(t)} r_a(t) \triangleq r_{a_*}(t)$. From the definition of arm $i_l^t$, its $l_{i_l^t}(t) \leq r_i(t)$. Using this and (7), we have that

$$G_i^r(l_{i_l^t}(t), t) = \min_{j:l_j(t)>l_{i_l^t}(t)} r_j(t) - l_{i_l^t}(t) \leq \min_{j:l_j(t)>r_i(t)} r_j(t) - l_{i_l^t}(t) = r_{a_*}(t) - l_{i_l^t}(t). \quad (42)$$

From the definition of arm $i_r^t$, we have that $\mathtt{U}\Delta_i^r(t) = r_{i_r^t}(t) - l_{i_l^t}(t) \leq r_{a_*}(t) - l_{i_l^t}(t)$ as argued above. That implies $r_{a_*}(t) \geq r_{i_r^t}(t) > l_{i_r^t}(t)$, which contradicts the supposition. $\qquad \square$

**Lemma 11.** *At any time $t \geq t(i, 2\tau)$, all arms $j \in \mathcal{A}_i^0(t) \cup \mathcal{A}_i^1(t)$ are such that $r_j(t) - l_i(t) \leq \Delta_{\max}$.*

*Proof.* Consider an arm $j \in \mathcal{A}_i^0(t) \cup \mathcal{A}_i^1(t)$, then $j \notin \mathcal{A}_i^2(s)$ for all $s \leq t$ for otherwise that would contradict corollary 2. Thus $j \in \mathcal{A}_i^0(s) \cup \mathcal{A}_i^1(s)$ for all $s \leq t$.

By choice of $\tau$ we have that $r_i(t(i, \tau)) - l_i(t(i, \tau)) = 2c_\tau \leq \Delta_{\max}$ from (39). Hence if arm $j \in \mathcal{A}_i^0(s)$ for any $s \in [t(i, \tau), t(i, 2\tau)]$, we have that $r_j(s) - l_i(s) \leq r_i(s) - l_i(s) \overset{(a)}{\leq} r_i(t(i, \tau)) - l_i(t(i, \tau)) \leq \Delta_{\max}$, where inequality (a) is by Lemma 9.

Hence $j \in \mathcal{A}_i^1(s)$ for all $s \in [t(i, \tau), t(i, 2\tau)]$. If $\mathtt{U}\Delta_i^r(s)$ is the largest gap upper bound then $i_R^s \notin \mathcal{A}_i^0(s)$ by the above reasoning. Then Lemma 13 states that arm $j$ was sampled anytime arm $i$ was sampled between $t(i, \tau)$ to $t(i, 2\tau)$. This implies that $T_j(t(i, 2\tau)) \geq \tau$, and we argue that $r_j(t(i, 2\tau)) - l_i(t(i, 2\tau)) \leq \Delta_{\max}$ in the following manner. The arm $j_*$ is the maximizer in (10).

**Case I:** $\mu_i < \mu_{j_*} < \mu_j$. Here we argue that $j \notin \mathcal{A}_i^1(t(i, 2\tau))$ because $l_j(t(i, 2\tau)) \geq r_i(t(i, 2\tau))$ as shown below.

$$\begin{aligned} l_j(t(i, 2\tau)) - r_i(t(i, 2\tau)) &\geq \mu_j - 2c_{T_j(t(i,2\tau))} - (\mu_i + 2c_{T_i(t(i,2\tau))}) \quad \text{(Lemma 5)} \\ &\geq \mu_{j_*} - \mu_i - 4c_\tau \quad \text{(Assumption on means and monotonicity of } c(s)\text{)} \\ &\geq \mu_{j_*} - \mu_i - \Delta_{i,j_*} = 0. \quad \text{(Using (39))} \end{aligned}$$

**Case II:** $\max\{\mu_i, \mu_j\} < \mu_{j_*}$. Here we argue that $r_j(t(i, 2\tau)) - l_i(t(i, 2\tau)) \leq \Delta_{\max}$ as shown below.

$$\begin{aligned} r_j(t(i, 2\tau)) - l_i(t(i, 2\tau)) &\leq \mu_j + 2c_{T_j(t(i,2\tau))} - (\mu_i - 2c_{T_i(t(i,2\tau))}) \quad \text{(Lemma 5)} \\ &\leq \mu_{j_*} - \mu_i + 4c_\tau \quad \text{(Assumption on means and monotonicity of } c(s)\text{)} \\ &\leq \mu_{j_*} - \mu_i + \Delta_{\max} - \Delta_{i,j_*} \leq \Delta_{\max}. \quad \text{(Using (39))} \end{aligned}$$

$\qquad \square$

**Lemma 12.** *Suppose arm $i$ is sampled at time $t$ because $\mathtt{U}\Delta_i^r(t) = r_{i_r^t}(t) - l_{i_l^t}(t)$ is the largest gap upper bound. Consider an arm $j$ whose confidence bounds satisfy any one of the following conditions.*

　　*1. $l_j(t) < l_{i_l^t}(t) < r_j(t)$, or*

　　*2. $l_j(t) < r_{i_r^t}(t) < r_j(t)$.*

*Then* `MaxGapUCB` *samples arm $j$ as well at time $t$.*

*Proof.* Suppose arm $j$ satisfies condition (1). Consider the right gap of arm $j$, we have that $\mathtt{U}\Delta_j^r(t) \geq G_j^r(l_{i_l^t}(t), t)$. If the value of $G_i^r(l_{i_l^t}(t), t)$ is obtained by the first branch of (7), then the value of $G_j^r(l_{i_l^t}(t), t)$ is also given by its first branch. That implies $\mathtt{U}\Delta_i^r(t) = \mathtt{U}\Delta_j^r(t)$, and hence $j$ is sampled if $i$ is sampled. If $j = i_r^t$, by condition (1) we have that $l_{i_r^t}(t) < l_{i_l^t}(t)$, which implies that $G_i^r(l_{i_l^t}(t), t)$ is obtained by the second branch in (7). Hence for all arms $a \neq i_l^t$ we have $l_a(t) < l_{i_l^t}(t)$ and $r_{i_r^t}(t) = r_j(t) = \max_{a \neq i} r_a(t)$. Considering the left gap of arm $j$, since $\{a : r_a(t) < r_j(t)\} \neq \emptyset$,

$$G_j^l(r_j(t), t) = r_j(t) - \max_{a:r_a(t)<r_j(t)} l_a(t) = r_{i_r^t}(t) - l_{i_l^t}(t) = \mathtt{U}\Delta_i^r(t),$$

and arm $j$ is sampled if $i$ is sampled. Finally suppose the value of $G_i^r(l_{i_l^t}(t), t)$ is obtained by the second branch in (7), and $j \neq i_r^t$. Then

$$G_i^r(l_{i_l^t}(t), t) = \max_{a \neq i} r_a(t) - l_{i_l^t}(t) = r_{i_r^t}(t) - l_{i_l^t}(t),$$
$$G_j^r(l_{i_l^t}(t), t) = \max_{a \neq j} r_a(t) - l_{i_l^t}(t) = \max\{r_i(t), r_{i_r^t}(t)\} - l_{i_l^t}(t) = r_{i_r^t}(s) - l_{i_l^t}(t),$$

where the last equality is true because if not, then $\mathtt{U}\Delta_i^r(t) \geq G_j^r(l_{i_l^t}(t), t) > G_i^r(l_{i_l^t}(t), t) = \mathtt{U}\Delta_i^r(t)$, which contradicts the condition that $\mathtt{U}\Delta_i^r(t)$ is the largest. Hence arm $j$ is sampled if $i$ is sampled.

Suppose now that arm $j$ satisfies condition (2). We divide the proof of this part into two cases.

**Case I:** Suppose $r_{i_l^t}(t) > r_{i_r^t}(t)$.

If the arm $i_l^t \neq i$, then we show that $G_i^r(l_{i_l^t}(t), t)$ cannot be the largest gap upper bound. Consider the arm $a_* \triangleq \arg\max_{a:l_a(t)<l_{i_l}(t)} l_a(t)$, it satisfies $l_i(t) \leq l_{a_*}(t) < l_{i_l^t}(t)$. Then $G_i^r(l_{a_*}(t), t) = \min_{a:l_a(t)>l_{a_*}(t)} r_a(t) - l_{a_*}(t)$, where the first branch of (7) is active because of arm $i_l^t$. But

$$\min_{a:l_a(t)>l_{a_*}(t)} r_a(t) = \min\{r_{i_l^t}(t), \min_{a:l_a(t)>l_{i_l^t}(t)} r_a(t)\} = \min\{r_{i_l^t}(t), r_{i_r^t}(t)\} = r_{i_r^t}(t).$$

That would imply

$$G_i^r(l_{a_*}(t), t) = r_{i_r^t}(t) - l_{a_*}(t) > r_{i_r^t}(t) - l_{i_l^t}(t) = G_i^r(l_{i_l^t}(t), t),$$

which contradicts the identification of arm $i_l^t$ as the one giving the value of $\mathtt{U}\Delta_i^r(t)$. The case that remains is if the arm $i = i_l^t$. For this part consider the following two sub-cases:

**Sub-case Ia:** The set of arms $\{a : r_a(t) < r_{i_r^t}(t)\} = \emptyset$. Since the number of arms $K > 2$, the value $\max_{a \neq i} r_a(t) > r_{i_r^t}(t)$, and hence if $G_i^r(l_{i_r^t}(t), t) = r_{i_r^t}(t) - l_{i_l^t}(t)$, then it must be due to the first branch in (7). That implies $l_{i_r^t}(t) > l_{i_l^t}(t) = l_i(t)$. Then consider the left gap for arm $j$ that satisfies condition (2). Since the set $\{a : r_a(t) < r_{i_r^t}(t)\} = \emptyset$, we have

$$G_j^l(r_{i_r^t}(t), t) = r_{i_r^t}(t) - \min_{a \neq j} l_a(t) = r_{i_r^t}(t) - l_{i_l^t}(t) = \mathtt{U}\Delta_i^r(t),$$

which implies that arm $j$ will be sampled if $\mathtt{U}\Delta_i^r(t)$ is the largest.

**Sub-case Ib:** The set of arms $\{a : r_a(t) < r_{i_r^t}(t)\} \neq \emptyset$. Consider the arm $a_* \triangleq \arg\max_{a:l_a(t)<l_i(t)} l_a(t)$, the domain in the maximization is not empty because of the following. By the case assumption, there is an arm $a$ whose $r_a(t) < r_{i_r^t}(t)$. If the left bound of this arm $l_a(t) > l_i(t)$, then $l_i(t) < l_a(t) < r_a(t) < r_{i_r^t}(t)$, which contradicts the identification of arm $i_r^t$ for $G_i^r(l_i(t), t)$. Hence its left bound must satisfy $l_a(t) < l_i(t)$. Now consider the right gap of arm $a_*$ defined above. Since $l_i(t) > l_{a_*}(t)$, we have that $G_{a_*}^r(l_{a_*}(t), t) = \min_{a:l_a(t)>l_{a_*}(t)} r_a(t) - l_{a_*}(t)$. But

$$\min_{a:l_a(t)>l_{a_*}(t)} r_a(t) = \min\{r_i(t), \min_{a:l_a(t)>l_i(t)} r_a(t)\} = \min\{r_i(t), r_{i_r^t}(t)\} = r_{i_r^t}(t),$$

which implies that $G_{a_*}^r(l_{a_*}(t), t) = r_{i_r^t}(t) - l_{a_*}(t) > r_{i_r^t}(t) - l_i(t) = \mathtt{U}\Delta_i^r(t)$, which is a contradiction. We are left with the following Case II.

**Case II:** Suppose $r_{i_l^t}(t) < r_{i_r^t}(t)$.

Let $a_*$ be such that $l_{a_*}(t) \triangleq \max_{a:r_a(t)<r_{i_r^t}(t)} l_a(t)$. Then $l_{a_*}(t) \geq l_{i_l^t}(t)$. If the previous inequality is strict, then we have that

$$l_{i_l^t}(t) < l_{a_*}(t) < r_{a_*}(t) < r_{i_r^t}(t),$$

which contradicts the identification of arm $i_r^t$ as the one giving the value of $G_i^r(l_{i_l^t}(t), t)$. Hence we have that

$$G_j^l(r_{i_r^t}(t), t) = r_{i_r^t}(t) - \max_{a:r_a(t)<r_{i_r^t}(t)} l_a(t) = r_{i_r^t}(t) - l_{i_l^t}(t) = \mathtt{U}\Delta_i^r(t),$$

and arm $j$ is sampled if arm $i$ is sampled because of $\mathtt{U}\Delta_i^r(t)$. $\qquad\square$

**Lemma 13.** *Suppose arm $i$ is sampled at time $t$ when $\mathtt{U}\Delta_i^r(t) = r_{i_r^t}(t) - l_{i_l^t}(t)$. If $i_r^t \in \mathcal{A}_i^2(t)$ then all arms in the set $\mathcal{A}_i^1(t) \cup \mathcal{A}_i^2(t)$ are sampled by* MaxGapUCB. *If $i_r^t \in \mathcal{A}_i^1(t)$ then all arms in the set $\mathcal{A}_i^1(t)$ are sampled by* MaxGapUCB.

*Proof.* The qualifying condition states that the arm $i_r^t \in \mathcal{A}_i^1(t) \cup \mathcal{A}_i^2(t)$, hence from definitions (33), (34) we have that $r_{i_r^t}(t) \geq r_i(t)$. By definition (8) the arm $i_l^t$ is such that $l_{i_l^t}(t) \in [l_i(t), r_i(t)]$. We first argue that all arms in the set $\mathcal{A}_i^1(t)$ are sampled. For arm $j \in \mathcal{A}_i^1(t), r_i(t) \leq r_j(t)$. If $l_j(t) \leq l_{i_l^t}(t)$, arm $j$ satisfies condition (1) of Lemma 12 and hence is sampled if $\mathtt{U}\Delta_i^r(t)$ is the largest. If on the other hand $r_j(t) \geq r_{i_r^t}(t)$, then arm $j$ satisfies condition (2) of Lemma 12 and hence it is sampled if $i$ is sampled. The remaining case is if $l_{i_l^t}(t) < l_j(t) < r_j(t) < r_{i_r^t}(t)$, but that would contradict the identification of the arm $i_r^t$ for $\mathtt{U}\Delta_i^r(t)$.

Now suppose arm $i_r^t \in \mathcal{A}_i^2(t)$, what is left to prove is that all arms in the set $\mathcal{A}_i^2(t)$ are sampled. Since $i_r^t \in \{a : l_a(t) > r_i(t) > l_{i_l^t}(t)\}$, we have that

$$G_i^r(l_{i_l^t}(t), t) = \min_{j:l_j(t)>l_{i_l^t}(t)} r_j(t) - l_{i_l^t}(t) = r_{i_r^t}(t) - l_{i_l^t}(t) = \min_{j \in \mathcal{A}_i^2(t)} r_j(t) - l_{i_l^t}(t),$$

where the last equality is true because arm $i_r^t \in \mathcal{A}_i^2(t)$ satisfies $l_{i_l^t}(t) > r_i(t) \geq l_{i_l^t}(t)$. From definition (34), any $j \in \mathcal{A}_i^2(t)$ is such that $l_j(t) \leq r_{i_r^t}(t)$ and satisfies condition (2) of Lemma 12. Hence arm $j$ is sampled if $i$ is sampled because of its right gap. $\qquad\square$

# E  Details for Section 6: Proof of Lemma 1

**Lemma 1.** *Consider a model $\mathcal{B}$ with $K = 4$ normal distributions $\mathcal{P}_i = \mathcal{N}(\mu_i, 1)$, where*

$$\mu_4 = 0, \quad \mu_3 = \epsilon, \quad \mu_2 = \nu + 2\epsilon, \quad \mu_1 = 2\nu + 2\epsilon,$$

*for some $\nu \gg \epsilon > 0$. Then any algorithm that is correct with probability at least $1 - \delta$ must collect $\Omega(1/\epsilon^2)$ samples of arm 4 in expectation.*

*Proof.* The maximum gap in $\mathcal{B}$ is $\Delta_{\max} = \Delta_{3,2} = \nu + \epsilon$. Define an alternate bandit model $\mathcal{B}'$ with 4 normal distributions $\mathcal{P}_i' = \mathcal{N}(\mu_i', 1)$ where

$$\mu_i' = \mu_i \quad \forall i \neq 4, \qquad \mu_4' = 2.1\epsilon.$$

Note that the ordering of the means in $\mathcal{B}'$ does not follow the subscript indices, indeed $\mu_3' < \mu_4'$.

Figure 10: Changing the original bandit model $\mathcal{B}$ to $\mathcal{B}'$. $\mu_4$ is shifted to the right by $2.1\epsilon$. As a result, the maximum gap in $\mathcal{B}'$ is between green and purple.

The two measures are illustrated in Fig. 10. The maximum gap in $\mathcal{B}'$ is $\Delta_{\max}' = \Delta_{2,1}' = \nu$ and $\Delta_{3,2}'$ is no longer a valid gap between consecutive arms. Consider algorithm for identifying the maximum gap and let $\widehat{C}_1$ denote the top-cluster returned by the algorithm when it stops at time $\tau$. Let

$E = \{\widehat{C}_1 = \{1, 2\}\}$. Assume that $\mathbb{P}_{\mathcal{B}}(E) \geq 1 - \delta$ and $\mathbb{P}_{\mathcal{B}'}(E) \leq \delta$. Letting $d(\cdot)$ denote the binary relative entropy, Lemma 1 in Garivier and Kaufmann [10] implies that

$$\sum_{a=1}^{4} \mathbb{E}_{\mathcal{B}}[T_a(\tau)]\mathsf{KL}(\mathcal{P}_a, \mathcal{P}'_a) \geq d(\mathbb{P}_{\mathcal{B}}(E), \mathbb{P}_{\mathcal{B}'}(E)) \geq d(1 - \delta, \delta)$$

$$\implies \mathbb{E}_{\mathcal{B}}[T_4(\tau)](\mu_4 - \mu'_4)^2 \geq \log \frac{1}{2.4\delta} \implies \mathbb{E}_{\mathcal{B}}[T_4(\tau)] \geq \frac{1}{(2.1\epsilon)^2} \log \frac{1}{2.4\delta}.$$

Similarly, one can show that $\mathbb{E}_{\mathcal{B}}[T_1(\tau)] \geq 1/\epsilon^2$ by creating an alternative bandit instance $\mathcal{B}''$ identical to $\mathcal{B}$ except $\mu''_1 = 2\nu + 3.1\epsilon$. $\qquad\square$