[Reviews · NeurIPS 2019]

Reviewer 1



This paper introduces a new bandit problem where the goal is to find the largest gap between any two adjacent arms. It is framed similarly as the best-arm identification problem in the sense that the goal of the player aims to identify the max gap between two adjacent arms using as less samples as possible. The authors provide a lower bound on sample complexity for that setting. They also propose elimination and UCB-like algorithms for which they provide sample complexity analysis matching the lower bound. Strengths: [Originality] * The tackled setting is a new problem. * The algorithms are based on known bandit-strategies, but their application for solving the problem is new. [Quality] * The authors provide a lower bound on sample complexity for their new setting and matching upper bounds for the proposed algorithms. * The simulated experiment is useful for showing the sample complexity improvement of proposed algorithms compared with uniform sampling. [Clarity] * The authors provide a lot of intuition on their theoretical results, along with examples to understand their definitions and proofs. [Significance] * It is interesting how the authors transform the confidence intervals on arm means into confidence intervals on arm gaps. These theoretical perspectives (and Alg. 4) could be useful in other problems. Weaknesses: [Clarity] * What is the value of the c constant (MaxGapUCB algorithm) used in experiments? How was it determined? How does it impact the performance of MaxGapUCB? * The experiment results could be discussed more. For example, should we conclude from the Streetview experiment that MaxGapTop2UCB is better than the other ones? [Significance] * The real-world applications of this new problem setting are not clear. The authors mention applicability to sorting/ranking. It seems like this would require a recursive application of proposed algorithms to recover partial ordering. However, the procedure to find the upper bounds on gaps (Alg. 4) has complexity K^2, where K is the number of arms. How would that translate in computational complexity when solving a ranking problem? Minor details: * T_a(t) is used in Section 3.1, but only defined in Section 4. * The placement of Figure 2 is confusing. --------------------------------------------------------------------------- I have read the rebuttal. Though the theoretical contribution seems rather low given existing work on pure exploration, the authors have convinced me of the potential impacts of this work.

Reviewer 2



This problem studies a novel variation of pure-exploration in multi-armed bandits, where the objective is to discover the pair of adjacent arms with the maximum gap of expectations. The authors provide elimination and UCB-based algorithms and show their upper bounds. They also provide a minimax lower bound to show the optimality of the proposed algorithms. They conduct experiments on both synthetic and real world data sets to verify the effectiveness of their algorithms. The problem studied in this paper is a novel variation of MAB problem and has some meaningful real-world applications. It provides a preliminary theoretical foundation to efficient learning-to-rank tasks as well as approximate query processing tasks. The unique challenge in this problem is the unknown order of arms, which introduces a non-trivial twist that differs the problem from traditional top arm identification. On the other hand, the output only needs to find the maximum gap, which is easier than figuring out the full rank of all the arms. The proposed algorithms are presented in a clear way and seem quite intuitive. The algorithms generally maintain the confidence intervals of gaps between arms, as well as an active set of arms which are possibly related to the max gap. The arms in the active set will be sampled and played. The upper bound of the proposed algorithms is related to how close “the gap of a pair of arms” is to “the max gap”, as well as the gap between a pair of arms per se. The authors also provide an intuitive explanation of the upper bound. A more practical question here is that if there are multiple pairs of arms with similar gap values, the algorithm still needs to take a lot of samples to figure out. The authors may consider a PAC-setting to prevent this situation. I have minor concern on the experiment settings. The theoretical results presented are obtained in a fixed-confidence setting. However, the experiments shown seem to be evaluating performance in a fixed-budget setting. Is there a specific reason why a fixed-confidence experiment is not conducted? Another concern is: do authors have more real-world data set test cases other than the Borda one? Currently the authors primarily run experiments on only one single test case in real-world data. However, it would be interesting to see 1) what would be the hardness distribution in a real-world data set. 2) does the algorithm performance align well with the hardness value. Generally, I think the paper studies an interesting novel problem setting. The paper is well-written and provides a solid foundation for future studies. There are some minor issues with experiments, but overall the paper would still be a valuable contribution to the community. %===After rebuttal===% Thanks for the authors for their response. I think the figure illustrating hardness and #samples is useful and could be included in the paper. I remain my relatively positive opinion.

Reviewer 3



This paper considers a variant of pure-exploration bandit problems where the agent tries to divide the arms into two sets such that the highest mean of one set is smaller than the lowest mean of the other set, and the gap between these means is maximized. Algorithms based on elimination and UCB (with/without LCB) are proposed, and their high-probability sample complexity bounds are derived with a discussion on the lower bound in the fixed-confidence scenario. Simulation results are given for synthetic setting and a one based on a real-world dataset. One of the main concerns is that the motivation is quite unclear. In the proposed framework, a very large number of samples is required if the largest and the second largest gaps are close to each other, as illustrated in the discussion on the lower bound, even though the objective of the framework is just clustering. Therefore there should be detailed discussion on in what situation such an exact clustering becomes necessary. For example, in the simulation for the Streetview dataset, the resulting “correct” clusters seem quite unbalanced and I don’t understand why finding such clusters is essential. For the same reason, the discussion “Note that finding the safest image (best-arm identification) is hard as we need a lot of human responses to decide the larger mean between the two rightmost distributions” seems quite inappropriate since it seems to needs a lot of human responses to decide the larger gap between the highest and the second highest ones. Another concern is that the techniques used in this paper in the evaluation of the sample complexity seem to be a naive application of the original analysis of the base algorithms, successive elimination and LUCB algorithms. As a result, the derived bound is only order-optimal (with respect to Delta) and does not evaluate the expected number of samples even though the lower bound is given for the expected number of samples. Furthermore, the sample complexity bound is only correct with probability 1-delta, where delta is pre-specified by the required confidence level and cannot be controlled in the evaluation. There should be more modern analysis on the lower bound and matching upper bound following the line of work by Kaufmann for best arm identification, techniques of which seem also applicable to this setting. Section 1.1: The considered model must be clarified (Bernoulli, Normal or sub-Gaussian,...) to discuss confidence bounds. Algorithm 2: There seems to be no explanation on the choice of c, even though the algorithm with small c cannot be correct. Experiments: It is strange that the error probabilities are evaluated even though the theoretical framework considers the fixed-confidence setting. -------- Reply to the rebuttal: - choice of c The rebuttal clarified that c=5 is used in the experiments as in Jamieson+, but, unlike Jamieson+, there is no argument on what choice of c is theoretically guaranteed. In fact, c does not appear even once in the proof of Theorem 1. - "characterization of the "closest" alternate model does not hold in the MaxGap bandit problem" In general bandit problems, the lower bound is characterized by an optimization problem over a family of alternate models, and the closest model does not have to be explicitly available, though the resulting lower bound becomes not explicitly written. The analysis like those around Figure 8 will be useful (and interesting) to simplify the candidates of alternate models but the current result itself does not seem to be very useful because of the same drawback as LUCB as pointed out in the first review. The characteristic like "other arms may have to be sampled even if its own gap is small" also appears in other extensions such as linear bandits.

[Author Response · NeurIPS 2019]

Dear Reviewers, we thank you for appreciating our work and the constructive feedback, and address your concerns below.

Figure 1: Stopping time experiments

**Experiments**: (Rev. 1,2,3) We showed empirical mistake probability plots because in reality one cannot always wait for an algorithm to run until termination (similar concerns and plots are shown in Bubeck 2009, Jamieson 2013). We ran stopping time experiments and show our results in Fig. 1b,c, where we plot the empirical stopping time against the theoretical sample complexity (Thm. 2) for different arm configurations. We choose the arm configuration in Fig. 1a containing three unique gaps - a small gap $\Delta_3$ and two large gaps $\Delta_2 < \Delta_1 = \Delta_{\max}$. The hardness parameter is changed by increasing $\Delta_2$ and bringing it closer to $\Delta_1$. We see a linear relationship in Fig. 1b which suggests that the sample complexity expression in Thm. 2 is correct up to constants. In Fig. 1c we include random sampling and see that our adaptive algorithms require up to 5x fewer samples when run until completion. Fig. 1b suggests that UCB may outperform Top2UCB (Rev. 1). Fig. 1c shows that the adaptive algorithms are robust to the hardness of the problem and always outperform random sampling, and the gains increase with hardness (Rev. 2). We used a lower bound based stopping condition for Random, Elimination, Top2UCB, and set $c = 5$ in the UCB stopping condition (value of $c$ chosen empirically similar to Jamieson 2013).

**Motivation, Applications and Extensions**: The MaxGap bandit can be used to *efficiently* find the "good" set, where good is defined using the max-margin criterion instead of being pre-specified as in the top-$k$ best-arm problem. This is equivalent to clustering the arms into 2 sets, and just like for any clustering algorithm, one can construct adversarial distributions where the clustering returned by the algorithm is different from a desired clustering. The extensions identified to address these issues such as the PAC framework (Rev. 2,3), balanced or constrained clustering (Rev. 3), are challenging, and great avenues for future work. For example, the PAC framework would help in stopping early if two large gaps are within $\epsilon$ of each other. In best-arm identification, the PAC problem is solved by constructing a lower bound on the *mean* of *every* arm and stopping as soon as the difference between the highest upper bound and the lowest lower bound of the active set is less than $\epsilon$. In the MaxGap problem, it is difficult to construct a lower bound on the *gap* (as opposed to mean) of *every* arm due to interaction effects; we can only construct a single lower bound on the maximum gap, and this makes the PAC extension non-trivial. We would like to highlight though that with $\epsilon$-close large gaps, our algorithms correctly focus on the large gaps (Fig. 7), and hence stopping early and clustering according to the empirical means will yield a reasonable clustering. We can similarly discuss the complications with other extensions. As pointed out by Rev. 2, we believe this work can act as a solid foundation for this and other extensions such as constrained clustering, clustering with more than 2 clusters (Rev. 1), clustering without specifying number of clusters, and adaptive multi-dimensional clustering.

**Naive Application of Best-arm Analysis**: (Rev. 3) While the algorithms we propose are conceptually similar to existing bandit algorithms, the analysis of MaxGapUCB is far from a trivial application of the UCB analysis. On a high-level, in best-arm identification, the number of samples of a sub-optimal arm $i$ is bounded by observing that

$$\text{Arm } i \text{ is pulled} \Rightarrow \mu_i + 2c_{T_i(t)} \geq \hat{\mu}_i + c_{T_i(t)} \geq \hat{\mu}_{(1)} + c_{T_{(1)}(t)} \geq \mu_{(1)} \Rightarrow 2c_{T_i(t)} \geq \mu_{(1)} - \mu_i = \Delta_i. \quad (1)$$

The last inequality *directly* bounds the number of samples $T_i(t)$ of a sub-optimal arm $i$. In MaxGapUCB, the gap upper bound is obtained using the confidence intervals of two arms, and the fact that a sub-optimal gap $(i, j)$ has the highest gap-UCB implies that

$$(\mu_j + 2c_{T_j(t)}) - (\mu_i - 2c_{T_i(t)}) \geq (\hat{\mu}_j + c_{T_j(t)}) - (\hat{\mu}_i - 2c_{T_i(t)}) \geq \Delta_{\max} \Rightarrow 2(c_{T_j(t)} + c_{T_i(t)}) \geq \Delta_{\max} - \Delta_{ij}. \quad (2)$$

Thus unlike the reasoning in (1), the number of samples from arm $i$ is coupled to the number of samples from arm $j$, and $T_i(t) \to \infty$ if $j$ is not sampled enough. We show in our analysis that this cannot happen in MaxGapUCB. Furthermore, any arm $i$ is coupled with multiple other arms since the ordering of the arms is unknown, and may have to be sampled even if its own gap is small - a phenomenon absent in best-arm analysis because of the independence of the arm means. In our proof, we account for all samples of an arm by defining states the arm can belong to (called levels), and arguing about the confidence intervals of the arms in unison. Please refer to the outline of the proof in Fig. 8 and the complete analysis of MaxGapUCB on pages 16-22.

**Modern Analysis and Lower Bound**: (Rev. 3) Kaufmann's Track-and-Stop algorithm estimates the optimal proportion of samples needed from an arm based on the instance-dependent lower bound and uses it to guide their sampling strategy. In the lower bound, they obtain the "closest" alternate bandit model having a different best-arm by changing the means of just two arms (see proof of Lemma 3 in their paper). However that characterization of the "closest" alternate model does not hold in the MaxGap bandit problem, and obtaining an instance-dependent lower bound and an asymptotic expected sample complexity bound would require new lower bounding techniques.

[Meta-Review · NeurIPS 2019]

Although the problem is not properly motivated (some good examples in information retrieval and databases should be detailed), and despite the weaknesses of the results pointed in the reviews, the paper is clear and the originality of the work well established after the rebuttal. I thus recommend to accept it, and I encourage the authors to follow the suggestions of the reviews for the final version (if the paper is finally accepted) or before a re-submission (if it is not).